# Synthetic band-structure engineering in polariton crystals with non-Hermitian topological phases

L. Pickup [1,4], H. Sigurdsson [1,2,4], J. Ruostekoski [3] & P. G. Lagoudakis [1,2✉]

Synthetic crystal lattices provide ideal environments for simulating and exploring the band structure of solid-state materials in clean and controlled experimental settings. Physical realisations have, so far, dominantly focused on implementing irreversible patterning of the system, or interference techniques such as optical lattices of cold atoms. Here, we realise reprogrammable synthetic band-structure engineering in an all optical exciton-polariton lattice. We demonstrate polariton condensation into excited states of linear one-dimensional lattices, periodic rings, dimerised non-trivial topological phases, and defect modes utilising malleable optically imprinted non-Hermitian potential landscapes. The stable excited nature of the condensate lattice with strong interactions between sites results in an actively tuneable non-Hermitian analogue of the Su-Schrieffer-Heeger system.

[1] Department of Physics and Astronomy, University of Southampton, Southampton SO17 1BJ, UK. [2] Skolkovo Institute of Science and Technology, Novaya Street 100, Skolkovo 143025, Russian Federation. [3] Physics Department, Lancaster University, Lancaster LA1 4YB, UK. [4] These authors contributed equally: L. Pickup, H. Sigurdsson. ✉email: Pavlos.Lagoudakis@soton.ac.uk

Particles subjected to potential landscapes with discrete translational symmetries, whether natural or artificially made, exhibit bands of allowed energies corresponding to the quasimomentum of the crystal's Bloch states[1]. For instance, electronic band theory explains the difference between insulating and conducting phases of materials, as well as their optical properties. With advances in energy band synthesis in atomic systems (optical lattices) or photonic crystals, complicated yet meticulous lattice investigations are now possible including superfluid-to-Mott insulator phase transitions[2], networks of Josephson junctions[3], and solitonic excitations[4,5]. When the symmetry of a periodic structure is broken and/or boundaries are engineered in a desired way, there can arise defect states, surface states, and bound states in the continuum that do not dissipate energy into the surrounding environment. Advancements in photonics have allowed for the design and study of nearly lossless waveguides, filters, and splitters[6], with applications in communications and biomedicine. Recent developments have led to the study of topological states of matter in photonics[7] and separately in cold atoms[8,9].

One-dimensional (1D) crystals provide the simplest platform to study non-trivial topological phases, the prime example being the Su–Schrieffer–Heeger (SSH) model[10,11]. Today, the Zak phase (or the 1D topological winding number)[12] has been measured in a system of cold atoms[13], followed by the demonstration of adiabatic Thouless pumping[14], and an electronic topological superlattice[15]. Recently, non-Hermitian solid-state and photonic systems have attracted a huge interest in the study of out-of-equilibrium topological phases[16–21], dissipative quantum physics[22–24], and the advantageous effects of unbroken parity–time symmetry[25].

In the optical regime, a rapidly developing platform for the study of the above-mentioned phenomena are exciton–polaritons (from here on polaritons), realised in semiconductor microcavities. These hybrid light–matter quasi-particles are formed by the strong coupling of light confined in Fabry–Pérot microcavities and electronic transitions in embedded semiconductor slabs[26]. Their dissipative and out-of-equilibrium nature permits condensation into excited states[27–29] that still presents a non-trivial task for cold atoms in thermal equilibrium[30].

In polaritonic systems, there are two processes available to sculpt a crystal lattice. The most commonly applied process is through periodically patterning of the cavity mode and/or the intracavity quantum wells (QWs). This is typically achieved through patterned metallic deposition on top of the sample[27,29], etch and overgrowth patterning techniques[31], surface acoustic waves[32], or micro-structuring a sample into arrays of micropillars[33–35]. Linear features such as Dirac cones and flat bands have been demonstrated with polaritons utilising etched lattices in Lieb[34] and honeycomb[36] geometries with topological transport recently reported[35,37], as well as non-linear dynamics of bright gap solitons[38,39]. The other process utilises the matter component of polaritons to produce periodic potentials through many-body interactions. Similar to dipole moment-induced optical traps for cold atoms[40], or photorefractive crystals[41], one can design an all-optical potential landscape for polaritons by using non-resonant optical excitation beams to create reservoirs of excitons, which result in effective repulsive potentials due to polariton–exciton interactions[42–47].

In this article, we realise an all-optical, actively tunable band-structure engineering platform harnessing reprogrammable non-Hermitian potential landscapes that result from interparticle interactions. The platform is actively tunable due to the use of a spatial light modulator to spatially sculpt the non-resonant excitation beam and the resulting potential. The sample used is a non-patterned planar $2\lambda$ GaAs-based cavity containing eight 6-nm InGaAs QWs[48] (for more details, please see 'Methods'). Utilising this platform, we demonstrate a variety of band structure features including polariton condensation into high-symmetry points in arbitrarily excited energy bands of the resulting Bloch states. By dimerising the potential landscape, we experimentally realise an analogue of the topologically non-trivial SSH system, resulting in the formation of split energy band states. We determine through theoretical investigations that there is a $\pi$ change in the Zak phase (1D Berry phase) of the bands between the two choices of inversion symmetry points in the dimerised lattice. This confirms that our system experimentally provides a platform for studying non-trivial topology in non-Hermitian systems. Finally, by introducing local defects in the potentials periodicity, we demonstrate controllable highly localised defect-state condensation opening up possibilities to investigate analogues of bright and dark solitonic gap modes in strongly non-Hermitian lattices.

## Results

**Uniform 1D chains**. We start by considering 1D chains of narrow non-resonant Gaussian pumps (full-width at half-maximum $\approx 2\,\mu m$) exciting co-localised polariton condensates, where the inter-condensate separation is kept constant along the chain (see Fig. 1). The band structure along the lattice can be characterised via a single image of the dispersion (energy resolved $k$-space) providing that the chain is parallel to the entrance slit of the spectrometer. In Fig. 1, we show the experimental real-space and $k$-space photoluminescence (PL) distributions along with the corresponding dispersions for linear chains of eight polariton condensates with a lattice constant ($a$) of approximately 13 μm for Fig. 1b–d and 8.6 μm for Fig. 1e–g. It can be seen in Fig. 1d, g that condensate chains exhibit clear band structure in the their dispersions with dominant occupation at the high-symmetry points of their reduced Brillouin zone and all the repeated zones within the free polariton dispersion. These results evidence that polaritons, generated at the pump spots, sense the periodic nature of the potential, resulting in macroscopic coherent Bloch states and thus qualifying the technique even for relatively few pump cells. Furthermore, the energy band wherein the system condenses can be controlled by changing the separation between neighbouring condensates as is demonstrated in Fig. 1d, g, where we realise access to non-linear condensate dynamics in arbitrarily excited states through all-optical control.

We note the intricate Talbot interference patterns observed experimentally in the regions perpendicularly away from the chains, e.g. in Fig. 1b. Such patterns were previously demonstrated for polariton condensates using a chain of etched mesa traps[49] and demonstrate the ability of optically imprinted condensates with the concomitant potentials to achieve effects of etched/patterned systems. Moreover, polaritons condensing into the high-symmetry points of the lattice, observed also in refs. [27,29,31], can be intuitively understood from the fact that these Bloch modes have the strongest overlap with the gain (pump) region. The results are verified both through diagonalisation of the non-Hermitian Bloch problem and by numerically solving the driven-dissipative Gross–Pitaevskii equation describing a coherent macroscopic field of polaritons under pumping and dissipation (see Supplementary Notes 1 and 2).

**Topologically non-trivial band gap opening in 1D chains**. Figure 2 shows the experimental dispersions in Fig. 2a–e and real-space PL distributions in Fig. 2g–k for chains of eight condensates, demonstrating the splitting and periodic doubling of the band as the difference between the long ($a_l$) and short separation ($a_s$) is increased (panels a → e and g → l). For marginal

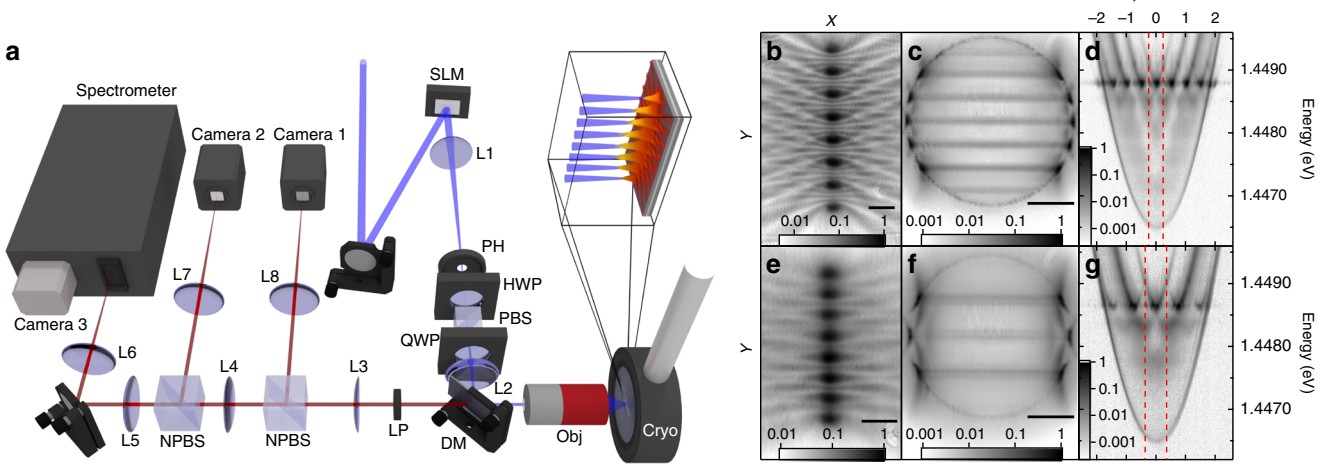

**Fig. 1 Experimental polariton condensate photoluminescence using eight non-resonant pump spots forming chains with uniform inter-condensate separation distances. a** Schematic representation of an experimental system where the blue beams represent the blue detuned non-resonant excitation laser and the red beams represent the photoluminescence. In the schematic: SLM spatial light modulator, PH pinhole, HWP half waveplate, QWP quarter waveplate, PBS polarising beam splitter, NPBS non-polarising beam splitter, DM dichroic mirror, obj microscope objective lens, cryo cold finger flow cryostat, LP long pass filter and L1–L8 planoconvex lenses. The inset of the schematic (top right) shows a zoomed in region of the sample at the focus of the sculpted non-resonant pump beam along with the resulting polariton distribution corresponding to **b**. **b–g** Logarithmic colourmaps showing the polariton condensate (**b**, **e**) real-space and (**c**, **f**) k-space distributions, and (**d**, **g**) the corresponding dispersions. The lattice constant is approximately 13 μm in **b–d** and approximately 8.6 μm in **e–g**. The black lines on the bottom right of **b**, **e** represent 15 μm and in **c**, **f** represent 1 μm⁻¹ scale bars. The red vertical dashed lines in **d**, **g** symbolise the boundaries of the reduced Brillouin zone of the polariton crystal.

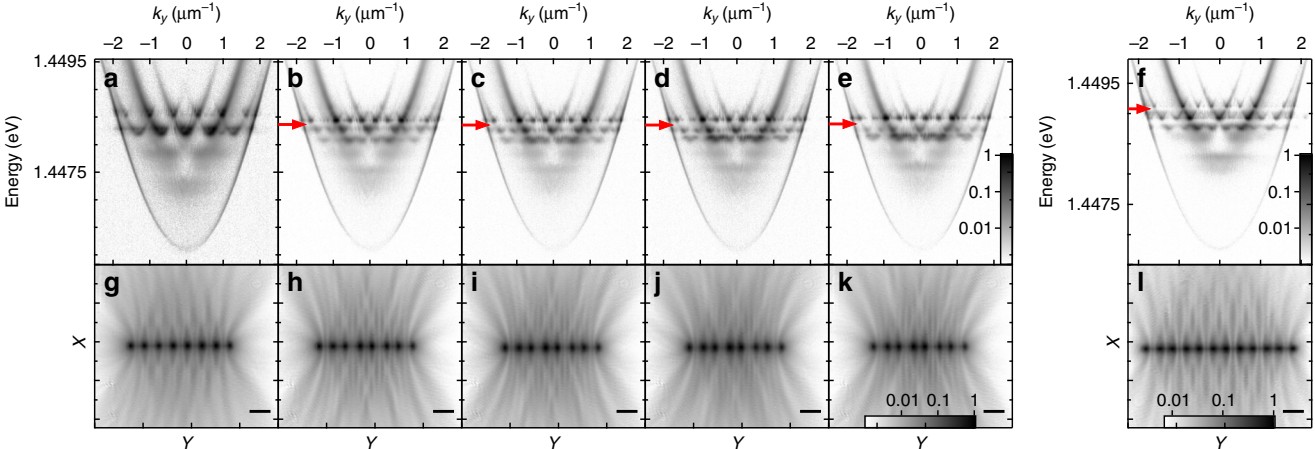

**Fig. 2 Experimental demonstration of a band gap opening as the difference between the long ($a_l$) and short ($a_s$) separation is increased.** Experimental dispersion (**a–f**) and corresponding real-space intensity distribution (**g–l**) of the PL from chains of eight polariton condensates with alternating separation distances where $a_l \approx 10.4$ μm and **a**, **g** $a_s \approx 10.4$ μm, **b**, **h** $a_s \approx 9.0$ μm, **c**, **i** $a_s \approx 8.9$ μm, **d**, **j** $a_s \approx 8.7$ μm, and **e**, **k** $a_s \approx 8.5$ μm. **f**, **l** shows the PL dispersion and real-space intensity distribution, respectively, for a chain of 12 condensates with $a_l \approx 10.2$ μm and $a_s \approx 9.2$ μm. The horizontal bars in the bottom right corner of each real-space distribution correspond to 15 μm and the red arrows indicate the gap opening in the dominantly occupied band. **a–e** are plotted using the logarithmic colour scales shown in **e** and **g–k** are plotted using the logarithmic colour scale shown in **k**.

differences in separation distance, $\delta = a_l - a_s$, the band gap formed is smaller than or comparable to the linewidths of the condensate polaritons and thus not fully resolvable. Increasing $\delta$ leads to an increased band splitting and the gaps become clearly visible when they exceed the polariton linewidth. In Fig. 2, the newly opened gap in the dominantly occupied energy band is indicated by the red arrows. Eventually for large enough $\delta$, the band splitting becomes significant enough that adjacent energy bands mix; see Fig. 2e. By increasing the number of unit cells in the experimental crystal potential, the splitting approximates the ideal infinite scenario (see the 'Methods' section for a discussion around the limits of the current experimental set-up). As a result,

the finesse of the band structure features becomes enhanced; this can be seen clearly in Fig. 2f, l, which show the dispersion and real-space distribution, respectively, of the PL from a chain of 12 condensates with $a_l = 10.2$ μm and $a_s = 9.2$ μm. We point out that in cold-atom systems topologically non-trivial band structures can be engineered by generating artificial gauge potentials using laser beams, where the hopping amplitude between adjacent lattice sites picks up a controllable phase factor (Peierls substitution) from the laser amplitudes[50–52] or from periodic modulation[53,54]. Here we have engineered an alternating pattern of tunnelling amplitudes between neighbouring polariton condensates by utilising the variation of the condensate hopping

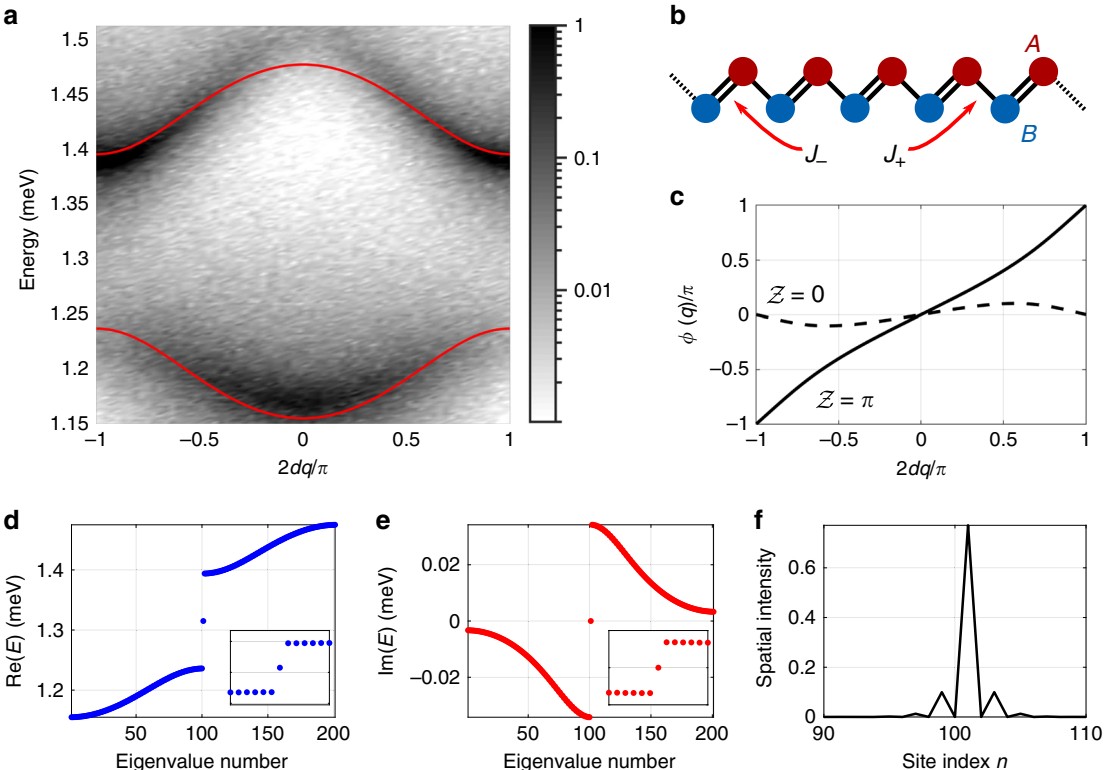

**Fig. 3 Simulation of the band gap formation for a staggered lattice and demonstration of a $\pi$ change in the Zak phase between the two choices of inversion centre. a** Greyscale colourmap showing the numerically time-resolved dispersion in a complex Gaussian potential lattice representing the experiment in Fig. 2. Zero energy represents bottom of the lower polariton dispersion and $2d$ is the lattice vector length. Red curves are calculated energies from Eq. (1). **b** A schematic illustrating the staggered lattice denoted by sublattice indices $A$ and $B$ and the two coupling strengths $J_{\pm}$. **c** Calculated $\phi(q)$ with solid and dashed curves corresponding to the two distinct centres of inversion in the chain by interchanging the values of $J_{\pm}$. **d, e** Real and imaginary eigenvalues from a finite system of Eq. (7) with 201 sites and a defect at site $|n = 101\rangle$ connected by two $J_+$ hoppings resulting in a midgap state at $E = \Omega$ (see insets). **f** Spatial density of the defect wavefunction. Edge sites are connected to the bulk by $J_-$ hoppings. Parameters are given in the 'Methods' section.

amplitude with the laser separation distance, such that interference of condensate polaritons between neighbouring sites is staggered.

The gain-localised nature of the condensate polaritons at their respective excitation spots permits description through discretised set of coherent polariton equations of motion. In particular, if the distances between adjacent condensates are weakly staggered the hopping amplitudes follow suit due to both differences in polariton travel times (i.e. the condensate envelope decays rapidly outward from its respective pump spot) and interference coming from their large outflow $k$-vector. The dimerised system, characterised by two distinct complex hopping amplitudes $J_{\pm}$, for long and short distance between the condensates, respectively, mimics a single-particle two-band problem representing a non-Hermitian version of the SSH model[10] (see 'Methods' and Fig. 3b). The single polariton Hamiltonian describing the two-sublattice chain in reciprocal space is written as,

$$\mathcal{H}(q) = \begin{pmatrix} \Omega & J_- + J_+ e^{iq} \\ J_- + J_+ e^{-iq} & \Omega \end{pmatrix}, \quad (1)$$

where $q$ is the crystal (Bloch) momentum and $\Omega$ is the on-site energy of polaritons at their pump spots. We note that $J_{\pm}$ are complex valued (see Eq. (6) in 'Methods'), but their conjugate is not taken in the lower off-diagonal element of the above Hamiltonian. This is due to the non-Hermitian nature of our system, which, in the context of topologically nontrivial phases, has taken a surge of interest[19–21,55–61]. In a ring-shaped lattice that forms periodic boundary conditions that we discuss later, the

Bloch waves are exact eigenstates and the description of the Zak phase also becomes exact.

The Bloch eigenstates belonging to Eq. (1) are written $|b^{(\pm)}\rangle = (\pm 1, e^{i\phi(q)})^T / \sqrt{2}$, where ($\pm$) denotes the upper (conduction) and lower (valence) band of the system. The energies belonging to these two bands are plotted as red curves in Fig. 3a in the first Brillouin zone. The standard procedure to validate the presence of topologically nontrivial phase transitions in 1D lattices is through the definition of the Zak phase[12], which can be regarded as the 1D parameter space extension of the geometric Berry phase,

$$\mathcal{Z} = i \int_{\text{BZ}} \langle b^\pm | \partial_q | b^\pm \rangle \, dq = -\frac{1}{2} \int_{\text{BZ}} \frac{\partial \phi(q)}{\partial q} \, dq. \quad (2)$$

The Zak phase can only take values 0 or $\pi$ (modulo $2\pi$) when the origin is chosen at an inversion centre of the system. By solving the eigenvalue problem posed by Eq. (1), the Zak phase can be calculated straightforwardly by integration over the Brillouin zone.

In Fig. 3, we present numerical results reproducing the experimental gap opening shown in Fig. 2f. Figure 3a shows the fitted gapped bulk dispersion from Eq. (1) (red curves) in the lattice Brillouin zone. The curves are plotted on top of a black-and-white colourmap showing the numerically time-resolved single-particle dispersion based on a Monte Carlo technique (see Supplementary Notes 1). Figure 3b shows a schematic of the staggered lattice. In Fig. 3c, we plot $\phi(q)$ across the Brillouin zone corresponding to the two distinct centres of inversion symmetry

in the dimerised lattice, which is the same as interchanging the values of $J_{\pm}$. Integrating $\phi(q)$ across the Brillouin zone reveals a $\pi$ change in the Zak phase between the dimerisations, marking the existence of two topologically distinct phases. The findings are corroborated through first-principle calculations on the polariton system Schrödinger equation (see Supplementary Notes 2). We point out that our system is very different from that of hybridised orbitals in micropillar chains[33], where in the current case, the opening of the gap arises from the staggered interference between adjacent polariton condensate 'antennas' (see Eq. (6)). Experimentally, the gap opening observed in Fig. 2 implies a topological phase transition due to the localisation of polariton modes at each pump spot. This is in analogy to deep periodic potentials where the particles occupy a single mode at each site in the lowest band (i.e. the wavefunction can be described as a superposition of localised Wannier functions). The strong non-Hermitian nature of our hybrid light–matter system instead opens new avenues towards topological physics where the localisation of the particles is not dictated by the potential minima of the lattice with evanescent tunnelling.

**Defect-state condensation.** Moreover, by optically engineering a defect state in the lattice, one can mimic the behaviour of solitons in the polyacetylene polymers of the original SSH model[10,11]. Such a defected system is depicted as …-B-A-B-A-B-A-A-B-A-B-A-B-… where one site is adjacent to either two short-distance or two long-distance neighbours. The generation of the SSH dimerisation and defect states here is analogous to the engineering of a controllable phase factor (Peierls substitution) for the hopping amplitudes between adjacent sites in cold-atom systems using laser-assisted tunnelling[62]. Solving the complex eigenenergies of a finite system (see Eq. (7)) including such a defect (e.g. one site linked by two $J_+$ couplings) one can observe in Fig. 3d–f that a defect (midgap) state forms in the system, clearly distinguished from the bulk as it lies at zero energy.

Broken translational symmetry in a uniform chain also results in gap (defect) states appearing. These manifest as dispersionless states in the band structure (indicated by the blue arrows in Figs. 4 and 5), showing strong spatial localisation around the position of the defect in the pump geometry. Figure 4 shows the experimental real-space PL distribution from a chain of 12 condensates with separation distances of $a \approx 10.2\,\mu m$ except between the central two pump spots where the separation is reduced to $a_d \approx 9.0\,\mu m$, creating a defect in the potentials periodicity. A corresponding gap mode is visible in the dispersion (indicated with the blue arrow in Fig. 4b) and the energy-resolved strip of real space (Fig. 4c) demonstrates strong spatial localisation of the condensate for the defect energy (Fig. 4e). Such strongly localised states could permit investigation into optically generated analogue of polariton bright gap solitons observed previously for polariton condensates in photonic lattices[39]. On the other hand, the delocalised band energetically above the defect state suffers significant suppression in condensate occupation spatially around the defect, representing a dark soliton-like mode (see Fig. 4d). This suppression is a consequence of the bulk energy bands vanishing around the defect and thus inhibiting energy flow between the left and the right bulk region of the optical polariton crystal. We present simulations on such defect states in Supplementary Notes 4.

Optically imprinting the potential landscape affords the ability to finely tune the spectral position of the defect state, within the gap, by only changing the defect length ($a_d$) in the excitation geometry. The PL dispersions for chains of 12 condensates with $a = 10\,\mu m$ for five defect lengths between $a_d = 8.9\,\mu m$ and $a_d = 7.1\,\mu m$ are shown in Fig. 5a–e. As the defect separation

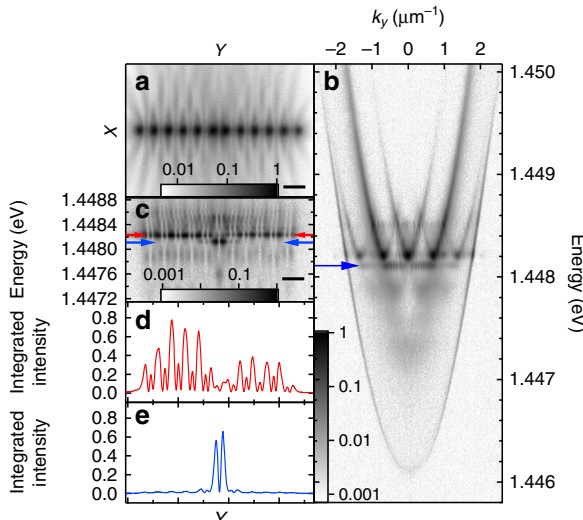

**Fig. 4 Experimental demonstration and characterisation of defect-state condensation in a chain of optically imprinted polariton condensates with an engineered defect.** Chain of 12 polariton condensates with $a \approx 10.2\,\mu m$ and $a_d \approx 9.0\,\mu m$. **a** Experimental real-space PL intensity distribution, **b** dispersion, **c** energy-resolved strip of real-space, **d, e** line profiles across energy-resolved real-space centred around the red and blue arrows in **c**, respectively. The horizontal solid black lines in **a**, **c** correspond to 15 μm and the normalised logarithmic colour scales are shown at the bottom of each colourmap. The gap mode is indicated by the blue arrow.

distance is reduced, the gap mode (indicated by blue arrows) blueshifts from the bottom of the gap to the top, at which point it begins to mix with neighbouring energy bands. For all excitation geometries shown in Fig. 5, the spatial distribution of the condensate occupying the defect state, and the energy band above it, have features comparable to those shown in Fig. 4d, e. We note that there also exists a dispersionless state in the next lower energy band gap that demonstrates the same blueshift behaviour with reducing defect length.

**Beyond finite 1D systems.** While the chains we investigate above show clear band formation with exquisite all-optical control over many band features including band splitting, dispersionless defect-state condensation, and arbitrarily excited band condensation, they remain finite systems. As shown in Fig. 2f, increasing the number of unit cells brings the system closer to the ideal infinite system and increases the fidelity of the band features. However, there are technical limitations to the size of chains that can be created, for example, due to the field of view of the objective or available power of the pump laser. In Fig. 6, we demonstrate polariton condensation in geometries of uniform and staggered octagons. Such a system implements a periodic boundary condition and provides a platform to avoid effects originating due to finite lattice sizes. Indeed, in ideal realisations of synthetic crystal lattices, one would like to achieve a well-defined crystal momentum for energy bands that follows from periodic boundary conditions. In optical lattices of cold atoms, such a system is difficult to create; the typical lattices have a finite length and they are additionally also superposed with a harmonic trapping potential. In a finite chain that we have considered until now, the description of eigenmodes in terms of their momenta is only approximate. To overcome this limitation, the ring-shaped lattice can be engineered for the polariton condensates in which case the Bloch waves of Eq. (1) form exact eigenstates of the corresponding tight-binding Hamiltonian Eq. (7) of the system. The presence of very weak radial modes clearly seen in

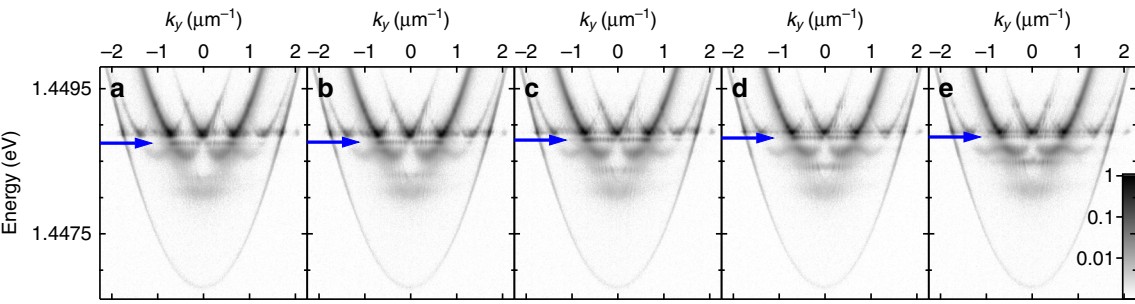

**Fig. 5 Demonstration of control over the energy of a defect state through changing the size of the engineered defect.** Experimental dispersions of the PL from chains of 12 polariton condensates with $a \approx 10\,\mu m$ and a defect length of **a** $a_d \approx 9.0\,\mu m$, **b** $a_d \approx 8.6\,\mu m$, **c** $a_d \approx 8.2\,\mu m$, **d** $a_d \approx 7.6\,\mu m$, and **e** $a_d \approx 7.1\,\mu m$. Each colourmap uses the normalised logarithmic colour scale shown in **e**. The gap mode is indicated by the blue arrows.

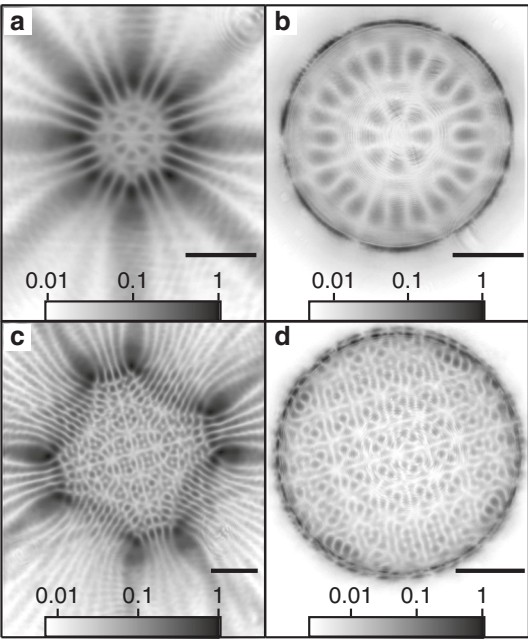

**Fig. 6 Experimental polariton condensate PL using eight pump spots forming a circle.** Logarithmic colourmaps showing the **a**, **c** real-space and **b**, **d** k-space photoluminescence distributions where **a**, **b** and **c**, **d** show regular and dimerised octagons, respectively. The black lines are 15 μm and $1\,\mu m^{-1}$ scale bars in **a**, **c** and **b**, **d**, respectively.

chains, the description of polariton Bloch eigenmodes in terms of their momenta is only approximate. Ring-shaped lattices, however, overcome such limitations where the definition of topological quantities like the Zak phase in the tight-binding limit becomes exact.

The observed defect-state condensation paves the way towards strong nonlinear lattice physics, with application in polaritonic devices such as information routing and fine tunable emission wavelength lasers. In addition, we expect that topological defect lasing can be realised by controlled defect preparation. We point out that the current study is performed in the scalar polariton regime but can be easily extended to include its spin degree of freedom by changing the polarisation of the pump, which creates different spin populations of the excitonic reservoirs feeding the condensates. Working with a horizontally polarised excitation, the system is chiral symmetric and each pump spot results in a randomly linearly polarised condensate. If interactions between the condensates, or on-site energies, are made spin dependent through typical photonic TE-TM microcavity splitting[63], or sample birefringence, then one gains access to spin-dependent band structures. This broadens the impact of nonresonantly generated artificial polariton lattices and, in principle, permits design of optical Chern insulators given the inherent spin–orbit coupling of polaritons in conjunction with applied magnetic fields[35,37]. Another exciting area for future research is expanding to topologically protected transport states with investigation into robustness against engineered imperfections.

logarithmic colour scale in Fig. 6 can be minimised by increasing the polygon's size. As long as the general features of the couplings between the condensates can be approximated by the tight-binding model, the assumption of the periodic boundary conditions remains valid. We point out that for our detection set-up the extraction of polariton band features along the polygons circumcircle in Fig. 6 is currently not possible.

## Discussion

Our study advances the emulation of many different lattice structures using a recyclable, and optically reprogrammable, multi-purpose platform in the strong light–matter coupling regime. The controllable condensation into arbitrarily excited Bloch states of the system gives access to excited orbital many-particle dynamics, which previously have been difficult to reach in solid-state systems. In particular, we address the challenge of realising a condensate lattice with periodic boundary conditions, which, in general, is attractive for analytical considerations (Bose–Hubbard model on a ring), and more closely resembles classic band-structure models of solid-state physics. In finite

## Methods

**Sample and experimental techniques**. We use a planar distributed Bragg reflector microcavity with a $2\lambda$ GaAs-based cavity containing eight 6-nm InGaAs QWs organised in pairs at the three anti-nodal positions of the confined field, with an additional QW at the final node either side of the cavity[48]. The sample is cooled to ~6 K using a cold finger flow cryostat and is excited with a monomode continuous wave laser blue detuned energetically above the stop band to maximise coupling in efficiency. The laser is modulated in time into square wave packets with a frequency of 10 kHz and a duty cycle <5% to prevent sample heating, and we operate at ~50% above the excitation density required for formation of a macroscopic coherent single-particle state. The sample has a vacuum Rabi splitting ~8 meV[48] and the regions of the sample utilised have an exciton–photon detuning of ~−3.5 meV.

The spatial profile of the excitation beam is sculpted using a phase-only spatial light modulator to imprint a phase map so that, when the beam is focused via a 0.4 numerical aperture microscope objective lens, the desired real-space is projected onto the sample surface. The same objective lens is used to collect the PL, which is then directed into the detection set-up. By controlling the spatial intensity distribution of the non-resonant excitation beam, we imprint a reprogrammable potential landscape[43,44,46] without the need of irreversible engineering. In the relaxation process from a non-resonant optical injection of free charge carriers to the polariton condensate, an incoherent 'hot' excitonic reservoir is produced that feeds the condensate. This reservoir is co-localised with the non-resonant excitation beam(s) and due to the strong polariton–exciton interaction results in a potential hill for polaritons where the excitation density is high[42]. This method additionally enables the elimination of large inhomogeneities since each element of the potential lattice can be adjusted through the power or shape of its respective pump element, such that the system achieves a homogeneous crystal structure.

In the current experimental set-up, when using similar lattice constants to those used throughout the manuscript, the upper limit of condensates in a 1D chain is approximately 14. However, we highlight that this is not a fundamental limit of the experimental technique. By replacing a few optical components, such as the microscope objective lens, this number could be increased. Equally by reducing the lattice constant, one can fit more non-resonant excitation beams. We note here that the lower limit of the lattice constant is determined by the width of the condensate bright centres, which approximately coincide with the Gaussian form of the nonresonant beam. In order to avoid strong overlap between the condensate centres, they should be separated by more than the FWHM of the pump beam.

**Theory.** The single particle dynamics of planar cavity polaritons, occupying the lower polariton dispersion curve, can be described by a two-dimensional Schrödinger equation[26].

$$ih\frac{d\Psi}{dt} = \left[ -\frac{\hbar^2\nabla^2}{2\mu} + V(\mathbf{r}) - \frac{ih\gamma}{2} \right]\Psi.$$  (3)

Here $\mu$ is the polariton mass, $\gamma$ is their lifetime, and $V(\mathbf{r})$ is the pump-induced complex potential. For the non-Hermitian lattice of Gaussian potentials, the interaction between polariton wavefunctions, gain-localised at their respective potentials, and separated by a distance $|\mathbf{r}_n - \mathbf{r}_m| = d_{nm}$, we can project the system onto an appropriate basis of wavefunctions. Omitting the diffusion of polaritons perpendicular from the chain, we consider a 1D system with the ansatz $\phi_n(x) = \sqrt{\kappa}e^{ik|x-x_n|}$, where $k = k_c + i\kappa$. The condensate wavefunction is then written,

$$\Psi(x,t) = \sum_n c_n(t)\phi_n(x).$$  (4)

Here $k_c, \kappa > 0$ represents the outflow momentum and decaying envelope of the polaritons generated at each potential. Given the narrow width of the pumps, we have approximated them as delta potentials, which, by direct integration, gives the following discretised single-particle equations of motion (details given in Supplementary Notes 2),

$$ih\frac{dc_n}{dt} = \Omega c_n + \sum_m J_{nm}c_m,$$  (5)

$$J_{nm} = \eta\left( V_0\cos(k_c d_{nm}) - \frac{\hbar^2 k_c}{\mu}\sin(k_c d_{nm}) \right)|H_0^{(1)}(k_c d_{nm})|.$$  (6)

Here $J_{nm}$ denotes the condensate hopping amplitudes, $\Omega$ is the complex-valued potential energy of polaritons generated at their respective pump spots, and $k_c$ is the outflow momentum of the polaritons from their pump spot, which depends on exciton–photon detuning, excitation beam waist, and excitation density[42]. $H_0^{(1)}$ is the zeroth order Hankel function of the first kind that accounts for the two-dimensional envelope of the propagating polaritons, $V_0 \in \mathbb{C}$ is the strength of the complex-valued pump-induced potential, and $\eta$ a fitting parameter. The physical meaning of Eqs. (5) and (6) is that condensate polaritons do not tunnel from one site to the next (evanescent coupling) but rather ballistically exchange energy. The term *ballistically* refers to the non-negligible polariton phase gradient away from the potentials determined by their strong outflow momentum $k_c$, which gives rise to the interferometric hopping dependence (sine and cosine functions).

In particular, in a distance staggered system (see Fig. 2) the condensates become linked by interchanging long and short distance $d_\pm = d \pm \delta$, respectively, where we assume $d \gg \delta$. For only nearest neighbour interactions, it leads to dimerisation of Eq. (5), which becomes characterised by two hopping amplitudes $J_\pm$. As a consequence, one obtains an approximate single-particle two-band problem representing a non-Hermitian version of the SSH model[10]. In the picture of second quantisation, Eq. (5) can be written as (see Supplementary Notes 2),

$$\begin{aligned}\mathcal{H} = {} & \Omega\sum_{m=1}^{M}\sum_\alpha |m,\alpha\rangle\langle m,\alpha| + J_+\sum_{m=1}^{M}(|m,B\rangle\langle m,A| + \text{h.c.}) \\ & + J_-\sum_{m=1}^{M-1}(|m+1,A\rangle\langle m,B| + \text{h.c.}).\end{aligned}$$  (7)

Here $|m,\alpha\rangle$ are state vectors of unit cell $m$ on sublattice $\alpha \in \{A,B\}$. With periodic boundary conditions, Eq. (7) can be diagonalised by standard Fourier transformation to the basis of crystal momentum $|q\rangle = M^{-1/2}\sum_{m=1}^{M}e^{imq}|m,\alpha\rangle$, where $q \in \{\delta_q, 2\delta_q, 3\delta_q, \ldots\}$ and $\delta_q = 2\pi/M$. It then follows that $\mathcal{H}(q) = \langle q|\mathcal{H}|q\rangle$ giving Eq. (1).

Parameters used for the calculations presented in Fig. 3 are: $d = 9.5\,\mu\text{m}$, $k_c = 1.5\,\mu\text{m}^{-1}$, $\mu = 0.32\,\text{meV ps}^2\,\mu\text{m}^{-2}$, $\eta = 0.24$, $\Omega = 1.315\,\text{meV}$, and $V_0 = 1.44 + i0.5\,\text{meV}$.

## Data availability
The data supporting the findings of this study are openly available from the University of Southampton repository at https://doi.org/10.5258/SOTON/D1194[64].

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

## Acknowledgements

We acknowledge the support of the UK's Engineering and Physical Sciences Research Council (grant EP/M025330/1 on Hybrid Polaritonics) and the RFBR project No. 20-52-12026 (jointly with DFG) and No. 20-02-00919. J.R. acknowledges the support of the UK's Engineering and Physical Sciences Research Council (grants EP/S002952/1 and EP/P026133/1).

## Author contributions

P.G.L. led the research project. P.G.L. and L.P designed the experiment. L.P. carried out the experiments and analysed the data. H.S. and J.R. developed the theoretical modelling. H.S performed numerical simulations. All authors contributed to the writing of the manuscript.

## Competing interests

The authors declare no competing interests.
