## [Peer Review File · Nature Communications]

Reviewers' comments:

Reviewer #1 (Remarks to the Author):

This is a very good paper and I recommend publication in Nature Communications. The authors perform an experiment on realising synthetic band-structure engineering in an optical exciton-polariton lattice. They demonstrate condensation into excited states of 1D lattices, rings, dimerised topological phases, etc. The stable nature of the condensate with strong interactions between the sites results in a tunable non-Hermitian analogue of the Su-Schrieffer-Heeger model. The authors also claim that extending to spin lattice physics, one can design optical Chern insulators given the inherent spin-orbit coupling of polaritons in conjunction with applied magnetic fields. It would be nice if the authors give a bit more details how the Chern insulators are designed.

Reviewer #2 (Remarks to the Author):

The manuscript by Pickup et al is a detailed study of polariton condensates generated in a GaAs-based microcavity by a quasi-continuous laser shaped into multiple excitation spots. Various excitation geometries are explored giving rise to non-trivial patterns, which are studied under the framework of simulation of condensed matter systems.

Broadly speaking, I believe this paper addresses novel issues that will be of interest in the community and also in other fields. This seems particularly true not only for the staggered excitation geometry, with its opened gaps and analogies with molecular Hamiltonians, but also for the condensation into gap states in lattice defects.

Moreover, the experimental techniques employed by the authors certainly make their study quite different to recent approaches to condensed matter simulators using polariton systems. In more detail, by using a sample with very low defect densities and yet relatively short polariton lifetimes, condensation is mostly restricted to the pump spot locations rather than away in reservoir traps. This also yields a unique promising platform for programmable simulators that is a lot more flexible than, for example, patterned microcavities.

Finally, a thorough theoretical analysis accompanies the paper, which can certainly help understand the complex non-equilibrium physics characteristic of polariton condensates.

However, before I can recommend this paper for publication, I believe there are a number of issues that must be resolved.

1) The paper organization needs strong revision. First, found it extremely hard to read the paper in a continuous flow. There are many cross references between main text and supplementary material, which in itself is poorly organized. For example, equations S13 and the one before it (which is not labelled) have quantities and refer to equations that are not presented till much later in the supp. info. (eqs. S31-S34). Also, even though Zak phases can be familiar for experts in the field, the fact that it's only defined in the supp. info. makes it hard to follow all the discussion around it in the main text for the first time a non-expert reads the paper.

Second, I do not understand what is the necessity to show any data nor simulations on the ring excitation geometry. The only consequence of it seems to be the authors' argument that this is important when periodic boundary conditions are relevant. Even though there is no doubt this is logically true, I believe there is nothing in the data to show that any of the lattice physics discussed in the paper is found in the ring data. For example, no gap is shown for the staggered ring excitation geometry (Figs. 1c,d and S2c,d) nor any defect in periodicity is explored. The way the ring excitation is presented, it simply shows that a laser beam can be arbitrarily shaped to create different excitation geometries, which is a pure optical table exercise, using a spatial light modulator in this case. The data simply shows polariton condensation in ring geometries, which

has already been shown by different research groups, including by some of the paper's authors.

2) I found the whole discussion around Zak phases to be a pure theoretical exercise. It does not seem to be necessary to invoke it to explain any of the experimental data. If that is the case, the abstract sentence "we demonstrate polariton condensation into [...] non-trivial topological phases" would be overstated. The same applies for the manuscript's title. Unless I am missing something, I would consider moving all discussions around Zak phases into the supp. info. or, alternatively, to a new paper.

3) It seems, from Fig. 3a-e, that the newly opened gap increases with increasing asymmetry in pump spot distance. Maybe the authors could extract that dependence, and compare it with what is expected from the SSH model? I understand the authors cannot predict the exact value of the inter-site coupling J , since Eq. 2 has free parameters. However it would be useful to check if they can fit a J vs d dependence. The authors themselves seemed to have some insights into the coupling amplitude between neighbouring condensates in one of their previous papers (Phys. Rev. X 6, 031032 (2016)).

4) Supp. info. S4 has a rich discussion around defect condensation. Interestingly, Fig. S6 shows how the total condensate density goes from being absent around the defect, to being confined to the defect, by simply changing the separation between pump spots at the defect location. The authors name these effects "dark-" and "bright gap solitons", respectively. The data in Fig. 6 in the main text shows the authors have done this exact experiment, i.e., have changed the distance between pump spots in the defect location, and have observed, in momentum space, the gap state shifting its energy, which agrees with the mentioned simulations. One therefore wonders if the total condensate density, in real space, agrees with the predicted bright/dark soliton physics found in simulations. It is particularly hard to compare intensities in Fig. 6 given that it is plotted in logarithmic scale. Could the authors show the corresponding real space experimental data, and its total integrated density, for the data presented in Fig. 6? This is particularly important since, in the last sentence on the introduction, the authors claim that "we demonstrate [...] analogues of bright and dark soliton modes". I have not seen such demonstration apart from numerical simulations.

5) All black-and-white images have a logarithmic color scale. However there are not ticks in the colorbars. Even though one understands that it is not necessary to show absolute intensity values, ticks are necessary in log plots to show the intensity ratio between two different colors.

6) I believe the it is not defined anywhere what the red arrows in Fig. 3 represent. (middle of the opened-gap?)

7) The simulated energy dispersions in Fig. S1i,l seem to show no particular higher density into a particular energy level, i.e., no condensation. Why is that so? Same question applies for Fig. S6c,f,i,l.

8) Fig. S4a,d: It seems the dotted lines refer to $\delta = 0$. If that's the case, could the authors precisely state that in the text?

Reviewer #3 (Remarks to the Author):

In this article, the authors propose a method to realize all-optical band structure engineering of exciton-polariton condensates using beam profile shaping with a spatial light modulator. Thanks to the repulsive interactions between the propagating condensate and the localized excitonic reservoir, non-resonant pump beams can be used to create optical potentials that can trap and manipulate the condensates. Here, using multiple beams the authors demonstrate condensate chains and rings with controllable spacing between the elements of the cells and show that by tailoring the potential landscape they can obtain non-trivial topologically states. As the paper is

currently written, it is unclear whether the authors are trying to highlight a new method or new physics of topological edge states. In the latter case, the novelty of the work is unclear since SSH and related models have already been realized and discussed in a variety of different systems (photonic crystals, metamaterials, plasmonic systems, polariton micropillars, etc.). The authors do not make clear how the results presented differ from these previous demonstrations and how the experiments advance the physical understanding of these systems.

Overall, the technique proposed in the paper offers increased flexibility in the design of complex potentials and is non-destructive, contrary to etching or patterning. However, using pump beams to create optical potential barriers for polaritons is not new, and the advance that the authors propose (using the spatial light modulator to imprint a more complex pattern than has previously been done) seems incremental, as presented. Additionally, if the focus of the paper is indeed the development of this new technique, then details are clearly lacking. The authors only give a bare-bone description of the experiment in the Method section, which isn't called anywhere in the main text. Similarly, sample information is also relegated to the Methods section (without being referred to in the results section) so that the reader doesn't get any information about the sample until they are done reading the main text.

In its current form, the paper is not ready for publication. The authors have clearly acquired a lot of data, but have not presented it in a clear manner, nor made substantive conclusions. This paper requires a substantial rewrite before publication can be considered. Further specific comments are below:

- 1) Who are the paper's target audience? There is currently very little background presented on optical manipulation of exciton-polariton condensates or on topological states and the Su-Schreiffer-Heeger model.
- 2) The authors write (p. 5): "[...] we start with a periodic ring-shaped arrangement of narrow Gaussian pumps (full-width-half-maximum $\approx 2 \mu\text{m}$) operating at 50% above the excitation density required for the formation of a macroscopic coherent single-particle state (see Figure 1)." Why 50%? What happens at higher or lower polariton densities? What is the exciton-photon detuning for these experiments? That affects the interaction strengths and thus, I would expect, the pump power needed to achieve the desired potential.
- 3) Figure 1 is very pretty but why is it there? I don't find it particularly illustrative of the experimental setup and the description is very brief with no discussion. Why is this geometry important? What do the changes in real and k-space mean?
- 4) The authors write (p. 5): "In cold-atom systems, periodic lattice potentials are typically superposed with harmonic trapping potentials, affecting the resulting finite band structure. Here, we engineer ring-shaped exciton-polariton crystals that more closely resemble ideal periodic bands of solid-state crystals, avoiding also any undesired edge effects appearing in finite linear condensate chains" but do not provide any explanations or context for readers who may not be familiar with such cold atoms experiments. How exactly does their setup differ from cold atom systems and why are these differences important? What are the "undesired edge effects" and how is their method removing them?
- 5) In the first few presented experiments, the authors use an eight condensate chain, then switch to twelve. Why? Some discussion of the effects of the effect of the number of pump beams/condensates on the experiments would be helpful. Similarly, I would have liked to read about the limits of the setup: What is the experimental limit on the number of cells that can be generated? The authors write (p.5): "[...] macroscopic coherent Bloch states even for relatively few pump cells, thus qualifying the technique." How few is too few? Is the minimum separation between condensates only limited by polariton linewidth? Other than lattice constant, can any other parameters be used to tune the hopping amplitude?

6) Page 9, the authors write: "We point out that even for a dimerised lattice ($\delta \neq 0$) the gap can be closed by only tuning which changes the interference between condensates" but do not explain how to tune k_c . Is it by changing the pump power? If it is, a discussion on how the pump power affects the system would also be useful.

7) Also page 9, the authors write: "We point out that our system is very different from that of hybridised orbitals in micropillar chains [31], where in the current case, the opening of the gap arises from the staggered interference between adjacent polariton condensate 'antennas'". I think this is an important point but the explanation is skimmed over. A stronger discussion of the physics behind the formation of the gap and how it differs from previous studies would have helped strengthen the paper.

8) The discussion section is rather superficial. What are other methods not good for realizing a lattice with periodic boundary conditions? If achieving a Bose-Hubbard model on a ring is important why was it so briefly discussed in the results section? How will changing lattice geometry give rise to topological defect lasing? How will spin be manipulated? Etc.

Minor comments:

Figure 2: Panels (b) and (e) are not commented on and don't seem to provide information that isn't more clearly illustrated in panels (a, c) and (d, f).

Figure 5: red and blue dashed lines in panel (c) are hard to see.

Page 9: "For the parameters given in the caption of Fig. 4 it closes when, e.g., $k_c \approx 1.395 \text{ } [\mu\text{m}]^{-1}$." The "e.g." confuses me. Are there multiple values of k_c that close the gap? If so, why and what are they?

Reviewers' comments:

Reviewer #1 (Remarks to the Author):

This is a very good paper and I recommend publication in Nature Communications. The authors perform an experiment on realising synthetic band-structure engineering in an optical exciton-polariton lattice. They demonstrate condensation into excited states of 1D lattices, rings, dimerised topological phases, etc. The stable nature of the condensate with strong interactions between the sites results in a tunable non-Hermitian analogue of the Su-Schrieffer-Heeger model.

The authors also claim that extending to spin lattice physics, one can design optical Chern insulators given the inherent spin-orbit coupling of polaritons in conjunction with applied magnetic fields. It would be nice if the authors give a bit more details how the Chern insulators are designed.

Reply:

We thank Reviewer 1 for the high appreciation of our work and recommendation for publication in Nature Communications. Inorganic semiconductor cavities are subject to splitting of the cavity TE and TM modes which gives rise to an effective spin-orbit coupling of polaritons known commonly as the *Optical Spin Hall Effect* [Leyder et al., Nature Physics **3**, 628 (2007)]. By designing a honeycomb lattice using our all-optical method and applying an external magnetic field one opens the gap between the spinor sublattice bands which, in honeycomb systems, give rise to topologically protected edge states [Nalitov et al., Phys. Rev. Lett. **114**, 116401 (2015)]. Furthermore, relevant to our study, recent works have focused on engineering artificial spin-orbit coupling and magnetic fields of polaritons by detailed alterations to the polariton potential geometry [Sala et al., Phys. Rev. X **5**, 011034 (2015); Jamadi et al., arXiv:2001.10395 (2020)]. We have now expanded the Discussion section to emphasize more on future works involving the spin structure of the polaritons and added references [34,36] in regards to Chern insulators.

Reviewer #2 (Remarks to the Author):

The manuscript by Pickup et al is a detailed study of polariton condensates generated in a GaAs-based microcavity by a quasi-continuous laser shaped into multiple excitation spots. Various excitation geometries are explored giving rise to non-trivial patterns, which are studied under the framework of simulation of condensed matter systems.

Broadly speaking, I believe this paper addresses novel issues that will be of interest in the community and also in other fields. This seems particularly true not only for the staggered excitation geometry, with its opened gaps and analogies with molecular Hamiltonians, but also for the condensation into gap states in lattice defects.

Moreover, the experimental techniques employed by the authors certainly make their study quite different to recent approaches to condensed matter simulators using polariton systems. In more detail, by using a sample with very low defect densities and yet relatively short polariton lifetimes, condensation is mostly restricted to the pump spot locations rather than away in reservoir traps. This also yields a unique promising platform for programmable simulators that is a lot more flexible than, for example, patterned microcavities.

Finally, a thorough theoretical analysis accompanies the paper, which can certainly help understand the complex non-equilibrium physics characteristic of polariton condensates.

However, before I can recommend this paper for publication, I believe there are a number of issues that must be resolved.

Reply:

We thank Reviewer 2 for appreciating our work and providing detailed constructive criticism.

Reviewer #2 (Continued):

1) The paper organization needs strong revision. First, found it extremely hard to read the paper in a continuous flow. There are many cross references between main text and supplementary material, which in itself is poorly organized. For example, equations S13 and the one before it (which is not labelled) have quantities and refer to equations that are not presented till much later in the supp. info. (eqs. S31-S34). Also, even though Zak phases can be familiar for experts in the field, the fact that it's only defined in the supp. info. makes it hard to follow all the discussion around it in the main text for the first time a non-expert reads the paper.

Second, I do not understand what is the necessity to show any data nor simulations on the ring excitation geometry. The only consequence of it seems to be the authors' argument that this is important when periodic boundary conditions are relevant. Even though there is no doubt this is logically true, I believe there is nothing in the data to show that any of the lattice physics discussed in the paper is found in the ring data. For example, no gap is shown for the staggered ring excitation geometry (Figs. 1c,d and S2c,d) nor any defect in periodicity is explored. The way the ring excitation is presented, it simply shows that a laser beam can be arbitrarily shaped to create different excitation geometries, which is a pure optical table exercise, using a spatial light modulator in this case. The data simply shows polariton condensation in ring geometries, which has already been shown by different research groups, including by some of the paper's authors.

Reply:

We have now added a label to the equation before Eq. (S13). We agree with Reviewer 2 that links between theory of the main text and the supplemental were not ideal and we have now taken considerable measures to amend this. We have now divided our Methods section into "Sample and experimental techniques" and "Theory". The latter contains information previously contained in the main text and the supplemental which we hope will make our study more coherent.

Furthermore, we have simplified the text surrounding Fig.3 [previously Fig.4] to make our theoretical approach to the system clearer to the readership. Description of the condensate dimerization is now described from a more intuitive perspective and we include now only two essential equations which define the single-particle band structure [Eq.(1)] and the definition of the Zak phase [Eq.(2)]. We also take care in guiding the reader to the Theory section where appropriate.

We appreciate the Reviewers remark that the periodic ring geometry is not ideal as the main message of the study. We have therefore modified Fig. 1 to now show data on linear chain of

condensates instead. Our data involving periodic condensates (i.e., polygons) has instead been moved to Fig. 6 at the back of the main text. We have now also included additional theoretical analysis which concerns Fig. 3 to make it more relevant to our study. This analysis concerns the existence of gap states in dimerised finite systems with a defect [see Fig.3(d,e)]. We therefore hope that by relocating our data on periodic systems, and bringing in more theoretical analysis, we have appropriately linked it more strongly to the manuscript.

However, we stress that Fig. 6 is not only to demonstrate the versatility of the experimental technique. In ideal realisations of synthetic crystal lattices, one would like to achieve a well-defined crystal momentum for energy bands that follows from periodic boundary conditions. In optical lattices of cold atoms such a system is difficult to create; the typical lattices have a finite length and they are additionally also superposed with a harmonic trapping potential. In a finite non-periodic chain of the polariton condensates the description of eigenmodes in terms of their momenta is only approximate. The ring-shaped lattice, however, overcomes all the limitations and we have crucially also realised it using the dimerised pattern of hopping amplitudes. This makes the Zak phase definition in the tight-binding limit exact.

We believe this is an important point to the broad readership of the journal interested in synthetic lattices and topological band structure in non-Hermitian systems.

Reviewer #2 (Continued):

2) I found the whole discussion around Zak phases to be a pure theoretical exercise. It does not seem to be necessary to invoke it to explain any of the experimental data. If that is the case, the abstract sentence "we demonstrate polariton condensation into [...] non-trivial topological phases" would be overstated. The same applies for the manuscript's title. Unless I am missing something, I would consider moving all discussions around Zak phases into the supp. info. or, alternatively, to a new paper.

Reply:

The discussion regarding the Zak phase is of high relevance to the system which we have studied. Designing artificial lattices which allow one to load particles into dimerized structures is currently a very hot topic, especially given the non-Hermitian nature of our system. For the past two years on topological phases in non-Hermitian systems see e.g., [Shen et al., Phys. Rev. Lett. **120**, 146402 (2018); Gong et al., Phys. Rev. X **8**, 031079 (2018); Lang et al., Phys. Rev. B **98**, 094307 (2018); Helbig et al., arXiv:1907.11562 [cond-mat.mes-hall] (2019); Hofmann et al., arXiv:1908.02759 [cond-mat.mes-hall] (2019); Lee et al., Phys. Rev. B **99**, 201103(R) (2019); Lee et al., arXiv:1912.06974 [cond-mat.mes-hall] (2020); Ghatak et al., J. Phys.: Condens. Matter **31**, 263001 (2019); Yoshida et al., Scientific Reports **9**, 16895 (2019); Yoshida et al., Phys. Rev. B **98**, 035141 (2018); Yoshida et al., Phys. Rev. B **99**, 121101(R) (2019); Banerjee et al., Phys. Rev. Lett. **124**, 063901 (2020); Borgnia et al., Phys. Rev. Lett. **124**, 056802, (2020)].

There are no easy direct detection methods of the topological invariant in 1D energy bands, and most experiments only perform indirect measurements (comparisons to the other parameters of the system, such as the lattice Hamiltonian terms, or the structure of the states and defects), which we also do here. We also do not think it is immediately obvious to the broad readership of the journal that energetic condensate polaritons traveling from one pump spot to the other make up a

dimerized system given the poorer exchange of energy between the spots when their relative distance increases. Typically, systems have considered evanescent coupling (tunnelling) between quantum wells, but the origin of the coupling is not what gives rise to nontrivial topology. As such, the dimerised ballistic coupling between the energetic condensates is a new method to explore topological physics at a very fundamental level which we aim at making clear to the reader.

Reviewer #2 (Continued):

3) It seems, from Fig. 3a-e, that the newly opened gap increases with increasing asymmetry in pump spot distance. Maybe the authors could extract that dependence, and compare it with what is expected from the SSH model? I understand the authors cannot predict the exact value of the inter-site coupling J , since Eq. 2 has free parameters. However it would be useful to check if they can fit a J vs d dependence. The authors themselves seemed to have some insights into the coupling amplitude between neighbouring condensates in one of their previous papers (Phys. Rev. X 6, 031032 (2016)).

Reply:

We thank the referee for an insightful comment. Originally we had considered comparing the gap opening for a weakly staggered lattice against the experimentally observed gap opening presented in Fig. 2. However, the staggering distances observed in Fig. 2 which allow us to confidently resolve the gap opening are beyond the weakly staggered approximation ($d \gg \delta$) of Eq. (7), which we emphasize now more clearly in the main text, and discuss in more detail in Section S2.3 in the SI. When the lattice is no longer weakly staggered one must apply continuum model to calculate the opening of the gap and the Bloch states of the system (which we have detailed in S2.1-S2.2 in the SI).

Reviewer #2 (Continued):

4) Supp. info. S4 has a rich discussion around defect condensation. Interestingly, Fig. S6 shows how the total condensate density goes from being absent around the defect, to being confined to the defect, by simply changing the separation between pump spots at the defect location. The authors name these effects "dark-" and "bright gap solitons", respectively. The data in Fig. 6 in the main text shows the authors have done this exact experiment, i.e., have changed the distance between pump spots in the defect location, and have observed, in momentum space, the gap state shifting its energy, which agrees with the mentioned simulations. One therefore wonders if the total condensate density, in real space, agrees with the predicted bright/dark soliton physics found in simulations. It is particularly hard to compare intensities in Fig. 6 given that it is plotted in logarithmic scale. Could the authors show the corresponding real space experimental data, and its total integrated density, for the data presented in Fig. 6? This is particularly important since, in the last sentence on the introduction, the authors claim that "we demonstrate [...] analogues of bright and dark soliton modes". I have not seen such demonstration apart from numerical simulations.

Reply:

We highlight here that the dispersion relations shown in Fig. S5 (previously Fig. S6) are single particle

dispersions relating to the corresponding potential landscape and do not represent the frequency components belonging to the simulated condensates shown in Fig. S5(b,e,h,k) and (d,g,j,m). Rather, these plots illustrate what the accessible dispersion looks like to polaritons prior to condensation. Our calculations are performed for low pump powers such that nonlinear renormalization of the single-particle dispersion due to the presence of the condensate is negligible.

The numerically simulated condensates from which spatial intensity distribution Fig. S5(b,e,h,k) and spectral intensities are extracted in Fig. S5(d,g,j,m) are single frequency states (i.e., stationary solutions of the driven-dissipative Gross-Pitaevskii equation), which is the reason why only a single spectral peak is seen in Fig. S5(d,g,j,m). When the lattice is weakly defected, the obtained stationary condensate solution [Fig.S5(e)] corresponding to the 'dark soliton' has a spectral peak coinciding with the upper bulk band [see Fig. S5(g)]. However, for moderately defected lattice, the stationary condensate spectral peak suddenly becomes degenerate with the defect state in the dispersion [see Fig. S5(j)] and one gets the 'bright soliton' mode shown in Fig. S5(h). We have now modified the text around Fig. S5 to highlight this more clearly.

In experiment the condensate can be characterized by several distinct spectral peaks (i.e. it is multimodal). This is due to the spontaneous scattering events of non-condensed polaritons which can seed multiple condensate modes. As such, in Fig. 4(c) [previously Fig. 5(c)] we conducted energy resolved real space imaging to be able to explicitly measure the spatial distribution of PL for different energies. We observe that the condensate mostly occupies two frequencies, indicated by the red and the blue arrows in Fig. 4(c) which correspond to the defect (blue) and the upper bulk band (red). When considering the upper bulk band, we experimentally find a strong suppression around the defect shown in linearly scaled line profile Fig. 4(d). This is comparable to the dark solitonic mode shown in Fig. S5(e). Similarly, the spatial PL at the defect state energy demonstrates strong localisation around the defect and notable suppression everywhere else as shown in Fig.4(e). This corresponds to the bright solitonic mode shown in S5(h). Such description of these states is included in the paragraph describing the results in Fig. 4 on p.10.

To summarise, the bright and dark soliton modes are explicitly demonstrated in Fig. 4. However, due to the multimodal nature of the condensate, the total real space PL investigated in Fig. 4 and Fig. 5 (previously Fig. 6) is in a superposition of the stationary condensate states shown in Fig. S5. When the experimental spatial distribution of the PL is energy resolved, as done in Fig. 4, we observe the individual soliton states comparable to those predicted in Fig. S5. We point out that Fig. 5 describes how the spectral position of the defect state can be controlled via the excitation geometry. For every panel in Fig. 5 there exist both the bright and dark solitonic modes energetically split from one another just as in Fig. 4. These line profiles are not shown explicitly in the manuscript as the general features are the same as for Fig. 4(d,e). The description in the main text regarding Fig. 4 and 5 has been expanded to clarify the states.

Reviewer #2 (Continued):

5) All black-and-white images have a logarithmic color scale. However there are not ticks in the colorbars. Even though one understands that it is not necessary to show absolute intensity values, ticks are necessary in log plots to show the intensity ratio between two different colors.

Reply:

We thank Reviewer 2 for highlighting that the ticks were missing on the colourbars, we have remade the figures ensuring that the ticks are now present.

Reviewer #2 (Continued):

6) I believe the it is not defined anywhere what the red arrows in Fig. 3 represent. (middle of the opened-gap?)

Reply:

Again we thank Reviewer 2 for bringing this to our attention. The caption of Fig. 2 (previously Fig.3) has been amended to indicate that the red arrows indicate the gap opened by staggering the potential and it is now also mentioned in the main text.

Reviewer #2 (Continued):

7) The simulated energy dispersions in Fig. S1i,l seem to show no particular higher density into a particular energy level, i.e., no condensation. Why is that so? Same question applies for Fig. S6c,f,i,l.

Reply:

The simulated energy dispersions in Fig. S1(i,l) are calculated using only the single-particle (linear) part of the Gross-Pitaevskii equation ($|\Psi|^2 \simeq 0$) by averaging over stochastic realizations of weak polariton fields [see Eq. (S7)]. The reason we display single-particle dispersion curves in Fig. S1(i,l), as opposed to the dispersion containing the frequency components of the condensate ($|\Psi|^2 \neq 0$), is to illustrate more clearly the bands belonging to non-condensed particles before condensation occurs. Indeed, due to the dominant Fourier components of the condensate the colourscale in Fig. S1(i,l) saturates and it becomes very difficult to resolve the surrounding dispersion. For this reason we plot the single-particle dispersion in Fig. S1 without the dominant condensate Fourier components. The same goes for Fig. S5 (previously Fig. S6). We now stress this more clearly in the text around Fig. S1 and Fig. S5.

Reviewer #2 (Continued):

8) Fig. S4a,d: It seems the dotted lines refer to $\delta = 0$. If that's the case, could the authors precisely state that in the text?

Reply:

We thank the referee for this important remark. The unit cell size is 19 microns, not 20 microns as previously written and the dotted lines correspond to $\delta = 0.5$ microns. We have clarified this in the

surrounding text and the caption of Fig.S4, and emphasize more clearly the difference between the solid and the dotted lines.

Reviewer #3 (Remarks to the Author):

In this article, the authors propose a method to realize all-optical band structure engineering of exciton-polariton condensates using beam profile shaping with a spatial light modulator. Thanks to the repulsive interactions between the propagating condensate and the localized excitonic reservoir, non-resonant pump beams can be used to create optical potentials that can trap and manipulate the condensates. Here, using multiple beams the authors demonstrate condensate chains and rings with controllable spacing between the elements of the cells and show that by tailoring the potential landscape they can obtain non-trivial topologically states. As the paper is currently written, it is unclear whether the authors are trying to highlight a new method or new physics of topological edge states. In the latter case, the novelty of the work is unclear since SSH and related models have already been realized and discussed in a variety of different systems (photonic crystals, metamaterials, plasmonic systems, polariton micropillars, etc.). The authors do not make clear how the results presented differ from these previous demonstrations and how the experiments advance the physical understanding of these systems.

Overall, the technique proposed in the paper offers increased flexibility in the design of complex potentials and is non-destructive, contrary to etching or patterning. However, using pump beams to create optical potential barriers for polaritons is not new, and the advance that the authors propose (using the spatial light modulator to imprint a more complex pattern than has previously been done) seems incremental, as presented. Additionally, if the focus of the paper is indeed the development of this new technique, then details are clearly lacking. The authors only give a bare-bone description of the experiment in the Method section, which isn't called anywhere in the main text. Similarly, sample information is also relegated to the Methods section (without being referred to in the results section) so that the reader doesn't get any information about the sample until they are done reading the main text.

In its current form, the paper is not ready for publication. The authors have clearly acquired a lot of data, but have not presented it in a clear manner, nor made substantive conclusions. This paper requires a substantial rewrite before publication can be considered. Further specific comments are below:

Reply:

We thank Reviewer 3 for her/his critical and insightful remarks on our work.

Methods to both design and control artificial lattices for simulation of solid state and molecular systems at their fundamental level has a very broad impact on the scientific community as is evident from the number of activities focused on photonic crystals, nanopatterned heterostructures, and optical lattices for atoms. In many of these systems the microscopic physics is very different, but underlying topologically non-trivial nature of the system is universal. The ultimate goal is to possess control over the full parameter space characterizing a lattice: Site depth (potential strength), lattice constant, number of sites, and hopping amplitude. The non-trivial topology itself also alters the

physics of these microscopically different realisations in fascinating ways that has potential applications ranging from topologically protected quantum information processing, unidirectional light transport, topological lasing, to name just a few.

With the increasing interest in non-Hermitian lattice physics, see [Shen et al., Phys. Rev. Lett. **120**, 146402 (2018); Gong et al., Phys. Rev. X **8**, 031079 (2018); Lang et al., Phys. Rev. B **98**, 094307 (2018); Helbig et al., arXiv:1907.11562 [cond-mat.mes-hall] (2019); Hofmann et al., arXiv:1908.02759 [cond-mat.mes-hall] (2019); Lee et al., Phys. Rev. B **99**, 201103(R) (2019); Lee et al., arXiv:1912.06974 [cond-mat.mes-hall] (2020); Ghatak et al., J. Phys.: Condens. Matter **31**, 263001 (2019); Yoshida et al., Scientific Reports **9**, 16895 (2019); Yoshida et al., Phys. Rev. B **98**, 035141 (2018); Yoshida et al., Phys. Rev. B **99**, 121101(R) (2019); Banerjee et al., Phys. Rev. Lett. **124**, 063901 (2020); Borgnia et al., Phys. Rev. Lett. **124**, 056802, (2020)], the parameter space now must be extended to complex valued hoppings and on-site potentials. In our study, we have demonstrated that this is possible by providing experimental evidence of a crystal band formation, controlled gap opening, defect states, and with an extensive theoretical analysis on the existence of topological phases in such driven-dissipative system of exciton polaritons. Importantly, our method of designing the lattice is non-invasive (i.e., no coherence is transferred from the beam to the polariton) and reprogrammable.

We now include a brief description of the sample used in the main text with a reference to a detailed experimental characterisation of the sample [Cilibrizzi et al. Appl. Phys. Lett. **105** 191118 (2014)]. Additionally, we now reference to the new subsection within Methods “Sample and experimental techniques” at the end of the introduction and have added an experimental setup schematic to Fig. 1 in the main text to elucidate the general methodology implemented.

We hope that with our point-by-point answers below and amended manuscript, we have adequately addressed the concerns of the reviewer.

Reviewer #3 (Continued):

1) Who are the paper’s target audience? There is currently very little background presented on optical manipulation of exciton-polariton condensates or on topological states and the Su-Schreiffer-Heeger model.

Reply:

We have now added references [44] and [46] to the list of references, [41-46], which guides the reader to studies in the past 10 years involved in engineering polariton potential landscapes through nonresonant optical excitation. We have additionally updated Fig.1 in our manuscript, with it now illustrating our experimental setup in order to transfer more clearly to the overall scientific community what are the requirements and procedures of generating an optically imprinted polariton lattice.

We have changed Fig. 3 (previously Fig. 4) and the surrounding text to make the topological background of our theory clearer to the journal’s readership. The Methods section has also been divided into two parts with one focused on the sample and experimental techniques and the other on the more general theoretical framework of polariton systems. We have added Refs.[55-61] as sources to recent studies (no older than 2-3 years) on topological physics in non-Hermitian systems.

We have also added a highly cited (2262 times 10-Mar-20) review on SSH systems to strengthen the topological background of our work, A. J. Heeger, S. Kivelson, J. R. Schrieffer, and W. -P. Su, *Solitons in conducting polymers*, Rev. Mod. Phys. 60, 781.

In summary, the target audience includes those interested in designing and/or investigating artificial lattices which can be loaded with coherent particle ensembles. This permits investigation into non-Hermitian macroscopic crystal (Bloch) physics in both periodic and non-periodic structures, where in the former the definition of topological invariants such as the Zak phase is precise. The system is also of broad interest due to nonlinearity of polaritons which we have demonstrated can form stable lattice gap solitons.

Reviewer #3 (Continued):

2) The authors write (p. 5): “[...] we start with a periodic ring-shaped arrangement of narrow Gaussian pumps (full-width-half-maximum $\approx 2 \mu\text{m}$) operating at 50% above the excitation density required for the formation of a macroscopic coherent single-particle state (see Figure 1).” Why 50%? What happens at higher or lower polariton densities? What is the exciton-photon detuning for these experiments? That affects the interaction strengths and thus, I would expect, the pump power needed to achieve the desired potential.

Reply:

The band structure features we observe are visible immediately from reaching threshold laser power and remain even notably above threshold. Operating a little above threshold results in a higher contrast of the features in the images captured without notable changes to the features themselves, this is why 50% above threshold was used.

The exciton-photon detunings for the region of sample used are around $\approx -3.5 \text{ meV}$ (now stated in the “Sample and experimental techniques” subsection). Whilst changing the detuning will change the threshold excitation power, the value stated is normalised to the corresponding threshold. Furthermore, the optical imprinting technique employed here allows us to freely adjust the potential landscape and therefore account for any changes in the coupling strength due to effects of parameters such as detuning.

Reviewer #3 (Continued):

3) Figure 1 is very pretty but why is it there? I don’t find it particularly illustrative of the experimental setup and the description is very brief with no discussion. Why is this geometry important? What do the changes in real and k-space mean?

Reply:

We refer to our answer to Reviewer 2, Comment 1, where the presence of data concerning periodic condensate geometries is clarified, and the adjustments we have made to the manuscript to make this data more streamlined with the study.

We have now updated Fig. 1 in our manuscript such that it illustrates our experimental setup in order to transfer more clearly to the overall scientific community what are the requirements and

procedures of generating a polariton condensate lattice. To clarify Fig. 6(a,b) (previously Fig.1(a,b)) show a uniform periodic condensate structure whereas Fig. 6(c,d) (previously Fig.1(c,d)) shows a dimerised periodic structure designed on the same sample.

Reviewer #3 (Continued):

4) The authors write (p. 5): “In cold-atom systems, periodic lattice potentials are typically superposed with harmonic trapping potentials, affecting the resulting finite band structure. Here, we engineer ring-shaped exciton-polariton crystals that more closely resemble ideal periodic bands of solid-state crystals, avoiding also any undesired edge effects appearing in finite linear condensate chains” but do not provide any explanations or context for readers who may not be familiar with such cold atoms experiments. How exactly does their setup differ from cold atom systems and why are these differences important? What are the “undesired edge effects” and how is their method removing them?

Reply:

We again refer to our answer to Reviewer 2, Comment 1. Additionally, we have now clarified and rewritten this part of the text which now reads:

“Indeed, in ideal realisations of synthetic crystal lattices, one would like to achieve a well-defined crystal momentum for energy bands that follows from periodic boundary conditions. In optical lattices of cold atoms such a system is difficult to create; the typical lattices have a finite length and they are additionally also superposed with a harmonic trapping potential. In a finite chain that we have considered until now, the description of eigenmodes in terms of their momenta is only approximate. To overcome this limitation, the ring-shaped lattice can be engineered for the polariton condensates in which case the Bloch waves of Eq.(1) form exact eigenstates of the corresponding tight-binding Hamiltonian Eq.(7) of the system.”

Reviewer #3 (Continued):

5) In the first few presented experiments, the authors use an eight condensate chain, then switch to twelve. Why? Some discussion of the effects of the effect of the number of pump beams/condensates on the experiments would be helpful. Similarly, I would have liked to read about the limits of the setup: What is the experimental limit on the number of cells that can be generated? The authors write (p.5): “[...] macroscopic coherent Bloch states even for relatively few pump cells, thus qualifying the technique.” How few is too few? Is the minimum separation between condensates only limited by polariton linewidth? Other than lattice constant, can any other parameters be used to tune the hopping amplitude?

Reply:

In the results presented in Fig. 1 and 2 (Previously Fig. 2 and 3) we use eight condensates as they yield clearly resolvable band features. As shown in Fig. 2(f), by increasing the number of condensates in the chain from eight to twelve we observe an increase in the finesse of the features and extinction in the gap, a consequence of reduced edge effects which lead to broadening of the bands. In an ideal

infinite system, the band linewidth only becomes limited by the cavity mirror quality (i.e., the cavity photon linewidth). We now stress the effects of increasing the number of unit cells more clearly at the bottom of p.5 and in the caption of Fig. 2.

“How few is too few?” is a good question but, to our knowledge, has a subjective opinion. Bloch momenta are not well defined in finite systems with no periodicity, but one can estimate when the band structure starts converging with increasing lattice size. In Fig.R1 we show the numerically resolved single-particle dispersion for an increasing number of pump potentials. The results show that around 8 pump spots the band features are already well defined, which reflects our choice of using 8 pump spots in experiment. We point out that, for example, in a recent paper of [Slot et al., *Nature Physics* **13**, 672–676 (2017)] only five lattice cells in the two orthogonal lattice directions were employed with good results. It is worth mentioning that this is precisely the importance of having access to periodic condensate systems [see Fig. 6 in main text]. There, the Bloch momenta is well defined and lattice quantities such as topological invariants are also well defined.

Figure R1: Numerically resolved dispersion of non-condensed polaritons for increasing number of Gaussian potentials (pumps).

The upper ‘limit’ of the setup was not discussed in the manuscript as any limit on this exact setup is far from a fundamental measurement of the limits of the technique. For example, by changing the microscope objective lens to one with a different field of view one would change the number of unit cells that could be imprinted. Such a discussion has been added to the “Sample and experimental techniques” section in Methods and is now referenced to in the main text.

The lower limit of the lattice constant is determined by the width of the condensate bright centers which approximately coincide with the Gaussian form of the beam, see e.g. [P. G. Lagoudakis & N. G. Berloff, *New J. Phys.* **19** 125008 (2017)]. In order to avoid strong overlap between the condensate centers, they should be separated by a distance at least greater than the FWHM of the pump beam. The non-resonant pump beams can be imprinted so that they overlap but this comes at the cost of having well defined condensate centres which permit discretisation of the problem. Indeed, if the pump spots are very close they will excite a single phase-uniform condensate and the concept of a

coherent polariton outflow (exchange of energy) from one pump spot to the next is not well defined. We have now added this explanation to the “Sample and experimental techniques” section.

The hopping amplitude is related to both k_c and d (the separation between nearest neighbour condensate centres). Therefore, in addition to changing the lattice constant one could equally change for example the detuning to adjust k_c . However, changing the detuning has a ‘global’ effect on the polariton crystal, i.e. it changes all the hopping amplitudes. One can tune k_c locally by adjusting the size of a given condensate (by changing the size of the corresponding non-resonant pump beam), but this also notably changes the local threshold condition. For localised changes to the hopping amplitude the most robust method thus far is keeping a constant FWHM of the pump beams across the lattice and just changing the separation distance.

Reviewer #3 (Continued):

6) Page 9, the authors write: “We point out that even for a dimerised lattice ($\delta \neq 0$) the gap can be closed by only tuning which changes the interference between condensates” but do not explain how to tune k_c . Is it by changing the pump power? If it is, a discussion on how the pump power affects the system would also be useful.

Reply:

Indeed, the outflow momentum of polaritons affects their ability to interact and is an important parameter. The outflow momentum can be tuned via exciton-photon detuning and excitation power [Wertz, E. Nat Phys **6** 860 (2010)] as well as the beam waist of the non-resonant excitation beam. We have now added this clarification below Eq. (1).

Additionally, since we do not demonstrate tuning of the condensate wavevector k_c , we have decided to remove the sentence:

“We point out that even for a dimerised lattice ($\delta \neq 0$) the gap can be closed by only tuning k_c which changes the interference between condensates. For the parameters given in the caption of Fig. 4 it closes when, e.g., $k_c \approx 1.395 \mu\text{m}^{-1}$.”

which we feel could mislead readers into thinking that such an experiment was performed.

Reviewer #3 (Continued):

7) Also page 9, the authors write: “We point out that our system is very different from that of hybridised orbitals in micropillar chains [31], where in the current case, the opening of the gap arises from the staggered interference between adjacent polariton condensate ‘antennas’”. I think this is an important point but the explanation is skimmed over. A stronger discussion of the physics behind the formation of the gap and how it differs from previous studies would have helped strengthen the paper.

Reply:

We have now modified the theoretical discussion around Fig. 3 (previously Fig. 4) and created a subsection within the Methods section titled “Theory” to make our discretized approach to ballistic polariton condensate clearer to the readership. We still refer to the supplementary information given the extensive analysis. Additionally, we have now modified the sentence,

“The term ballistically refers to the presence of the sine and cosine functions which originate from the interference of gain localised wavefunctions with outflow momentum k_c ”,

into,

“The physical meaning of Eq.(5)-(6) is that condensate polaritons do not tunnel from one site to the next (evanescent coupling) but rather ballistically exchange energy. The term ballistically refers to the non-negligible polariton phase gradient away from the potentials determined by their strong outflow momentum k_c which gives rise to the interferometric hopping dependence (sine and cosine functions)”.

We hope this will help the readers to appreciate the different physics of our system where polaritons are highly energetic as opposed to other system relying on coherent particle ensembles loaded into the low energy ground state of a given potential.

Reviewer #3 (Continued):

8) The discussion section is rather superficial. What are other methods not good for realizing a lattice with periodic boundary conditions? If achieving a Bose-Hubbard model on a ring is important why was it so briefly discussed in the results section? How will changing lattice geometry give rise to topological defect lasing? How will spin be manipulated? Etc.

Reply:

We have now made an alteration to the discussion section where we replace “overcome” by “address” in the following sentence:

“In particular, we address the challenge of realising a condensate lattice with periodic boundary conditions...”

The reason for our change is we do not want to convey the message that the realization of a periodic polariton condensate structure is a novel result. Other works have demonstrated such periodic systems in the context of polaritons, see e.g., [Sala et al., Phys. Rev. X **5**, 011034 (2015); Kalinin et al., arXiv:1710.03451 [cond-mat.mes-hall] (2017)]. We do however stress that the Bloch momenta are only well defined in periodic (or infinite) systems which is the crucial point of demonstrating a periodic geometry of polariton condensates, especially with a two-band sublattice structure such as the SSH model. This, to our knowledge, has not been considered before in terms of periodic polariton condensates. The periodic boundary conditions ensure that the Bloch momentum is a good quantum number and all lattice quantities, including topological invariants, are well defined.

We have now modified the Discussion on this topic so it now reads:

“In particular, we address the challenge of realising a condensate lattice with periodic boundary conditions which, in general, is attractive for analytical considerations (Bose-Hubbard model on a ring), and more closely resembles classic band-structure models of solid-state physics. In finite chains

the description of polariton Bloch eigenmodes in terms of their momenta is only approximate. Ring-shaped lattices, however, overcome such limitations where the definition of topological quantities like the Zak phase in the tight-binding limit becomes exact.”

We thank the reviewer for pointing out an unclear discussion element regarding defect state lasing. We stress that lasing (condensation) into topologically protected gap modes can be engineered by introducing a defect into bulk, as opposed to using *edge states* where the termination of the lattice gives rise to gap modes. We now write in the Discussion:

“Additionally, we expect that topological defect lasing can be realised by controlled defect preparation.”

The spin structure of the polariton can be manipulated by creating different spin populations of the excitonic reservoir feeding the condensate through the circular polarization of the nonresonant beam. Indeed, if working with a horizontally polarized excitation the system is chiral symmetric and each pump spot results in a randomly linearly polarized condensate. If interactions between the condensates, or their on-site energies, are made spin-dependent through typical photonic TE-TM microcavity splitting, or sample birefringence, then one gains access to spin-dependent bandstructures, which broadens the impact of nonresonantly generated artificial polariton lattices. We have integrated the above reply to the Discussion of our work.

Reviewer #3 (Continued):

Minor comments:

Figure 2: Panels (b) and (e) are not commented on and don't seem to provide information that isn't more clearly illustrated in panels (a, c) and (d, f).

Reply:

The k-space figures (now Fig. 1(b & e)) are included to explicitly demonstrate that all features of the band structure occur in one axis of k-space which then allows us to characterise the band structure through standard angle resolved spectroscopy. This has now been added to the main text explicitly in the first paragraph of the Results section.

Reviewer #3 (Continued):

Figure 5: red and blue dashed lines in panel (c) are hard to see.

Reply:

We agree with Reviewer 3 that the dashed lines in Fig. 5(c) (now Fig. 4(c)) were not so visible in printed versions of the manuscript. Therefore, we have replaced the dashed lines with coloured arrows to signify the energies for which the line profiles (Fig. 4(d & e)) are extracted whilst maintaining clarity of the features in Fig. 4(c).

Reviewer #3 (Continued):

Page 9: "For the parameters given in the caption of Fig. 4 it closes when, e.g., $k_c \approx 1.395 \text{ } [\mu\text{m}]^{-1}$." The "e.g." confuses me. Are there multiple values of k_c that close the gap? If so, why and what are they?

Reply:

We now clarify that our tight-binding treatment [Eq.(7)] is valid for small staggering of the lattice. Indeed, due to the sine and the cosine functions in Eq.(6) [previously Eq.(2)], arising from the strong interference effects of polaritons, the coupling has roots which limit the range of validity of Eq.(7). Beyond this approximation one needs to apply continuum Bloch theorem (which is done in S2 in the SI) to accurately describe the gap opening for increased staggering. The limit of the tight binding treatment is now discussed in more detail below Eq.(S27) in the Supplementary Information.

REVIEWER COMMENTS

Reviewer #1 (Remarks to the Author):

I am completely satisfied with the response of the authors to my and other Referee comments. In conclusion, I think that the revised version of the paper can be published.

Reviewer #2 (Remarks to the Author):

I have reviewed the new version of the manuscript by Pickup et al, as well as their responses to all reviewers. Overall, the authors have addressed most of the raised concerns, and as a result the manuscript quality has improved substantially. The paper is better structured and reads much better. The addition of an experiment setup image and a focus on the 1-dimensional data also helps to deliver the most important messages. As before, I still believe this paper is appropriate for a Nature Communications audience and presents enough novelty. However, there are still important open concerns that will prevent recommendation for publication at this stage. Those are:

1) The authors have chosen to keep the ring data in the paper. They argue the importance here are the periodic boundary conditions, and that it shows substantial advantages in comparison with cold atoms where a superimposed parabolic trap can't be avoided. While I don't doubt a circular geometry is indeed periodic along its circumference, and that polariton condensates can offer various advantages with respect to cold atoms, I suspect the emphasis the authors want to give to their data is out of place. In more detail, I believe the idea that a circular geometry is equivalent to a periodic 1D geometry is false for the data presented here. For one thing, polaritons in Figs 6a,c don't see a periodic potential, be it 1D (as in refs 27 and 31) or 2D (as in ref 29). Therefore, the sentence "polaritons condensing into the high symmetry points of the lattice, observed also in Refs. [27, 29, and 31], can be intuitively understood from the fact that these Bloch modes have the strongest overlap with the gain (pump) region" does not apply to the ring excitation geometry. In fact, in a ring excitation geometry polaritons from next-neighbours will not be blocked by the repulsive potential of each adjacent spot, and as a result all pump spots will be coupled to each other with different coupling strengths. If that's the case, the ring geometry is by no means equivalent to Fig. 3b with periodic boundary conditions. I have previously asked the authors about the appearance of gaps in the ring geometry. They have added a new sentence to the manuscript: "We point out that for our detection setup the extraction of polariton band features along the polygons circumcircle in Fig. 6 is not possible at current". While this is understandable from their new experiment setup figure (e.g. absence of a stepper motor to allow for a tomographic acquisition of 2-dimensional energy dependencies), I don't think a full band picture along the whole circumference is necessary to observe an energy gap. For example, one can simply acquire a 1-D energy dependence along a line intersecting two pump spots belonging to the ring, which should show a gap if the author's claims are true. According to the setup image, exciting with a linear or with a ring geometry does not change any of the acquisition part, so nothing prevents the authors from taking spectrometer data while changing their excitation profile. Can the authors provide such a data? Even if that data was not or cannot be taken, theoretical simulations should be able to clarify this discussion. Full 1-dimensional, or even full 2-dimensional, energy profiles can be produced with their simulated data. Could the authors provide such a simulation and, hopefully, show that a gap is indeed present?

2) The authors have kept the discussion about Zak phases in the main text, however in a simplified version to improve readability – which is much appreciated. However, they have failed to give the background of such a topic, and especially how it is important for the data described here. Stating that the concept is a "very hot topic" and giving numerous references to it does not help one understand the meaning of it. Therefore I must ask again: which role does Zak phases play into the data displayed here? Is it related to the opening of the gap, just from topological

continuity arguments? For example, will the size of the energy gap be given by a condition where the wavefunction phase undergoes a $0, 2\pi, 4\pi$ etc phase jump from one end to the other end of the Brillouin zone? Does it help to give topological protection to any of the observable quantities? I insist, the way the whole discussion around Zak phases is presented in the text seems to be a pure mathematical exercise. The text still fails to provide any intuition about how it influences the observed experimental data.

3) I thank the authors for explaining that the simulated dispersions are for single particles, therefore not to a condensate, and apologize for not having noticed it. However, I am now confused: Fig. S1i shows that the polariton single-particle dispersion reflects the lattice's periodicity, and Fig. S1l shows that dimerization opens up gaps in the single-particle dispersion. Also, according to Figs. S5c,f,i,l, soliton-like defect states also seem to be present in the single-particle physics they present. Therefore, one naturally wonders if condensation is necessary to observe any of the main features described in this paper, namely gaps and solitonic defects. In their response to referee #3, the authors seem to state that this is indeed the case: "The band structure features we observe are visible immediately from reaching threshold laser power and remain even notably above threshold", Could the authors present such a experimental data where, for the same pump spot geometry, just before condensation threshold no gaps nor solitonic features are present, however they appear above condensation? If that's the case, could the authors explain why, in response to referee #3, the authors say that "one-particle plots show what accessible dispersion looks like to polaritons prior to condensation"?

4) I thank the authors for adding ticks to the logarithmic colour scales. However, I must ask them to also add labels to the ticks in the main text data, or at least say in which basis the logarithmic scale is. As it is now, one cannot know if, e.g., the ratio between black and grey is 10, e, 2, 100, etc.

5) Apart from now stating that the exciton-photon detuning is -3.5meV , could they please also state what the sample's Rabi splitting detuning is, such that one knows the photonic/excitonic character of the presented data?

6) In the caption of Fig S1: "Panels (c,f)" should read "Panels (i,l)" instead.

Reviewer #3 (Remarks to the Author):

The authors have adequately answered all my questions and have significantly improved the manuscript from its initial version. Although some of the language is still a little awkward at times, I find the paper now much easier to read: the authors have reorganized the text in such a way as to make their goals and conclusions much clearer. The added method sections with the experimental details and theory is particularly helpful, in addition to the change in figure 1. In my opinion, the manuscript is now suitable for publication.

I have reviewed the new version of the manuscript by Pickup et al, as well as their responses to all reviewers. Overall, the authors have addressed most of the raised concerns, and as a result the manuscript quality has improved substantially. The paper is better structured and reads much better. The addition of an experiment setup image and a focus on the 1-dimensional data also helps to deliver the most important messages. As before, I still believe this paper is appropriate for a Nature Communications audience and presents enough novelty. However, there are still important open concerns that will prevent recommendation for publication at this stage. Those are:

We thank the reviewer for the constructive criticism provided which has improved the delivery of our study.

1) The authors have chosen to keep the ring data in the paper. They argue the importance here are the periodic boundary conditions, and that it shows substantial advantages in comparison with cold atoms where a superimposed parabolic trap can't be avoided. While I don't doubt a circular geometry is indeed periodic along its circumference, and that polariton condensates can offer various advantages with respect to cold atoms, I suspect the emphasis the authors want to give to their data is out of place. In more detail, I believe the idea that a circular geometry is equivalent to a periodic 1D geometry is false for the data presented here. For one thing, polaritons in Figs 6a,c don't see a periodic potential, be it 1D (as in refs 27 and 31) or 2D (as in ref 29). Therefore, the sentence "polaritons condensing into the high symmetry points of the lattice, observed also in Refs. [27, 29, and 31], can be intuitively understood from the fact that these Bloch modes have the strongest overlap with the gain (pump) region" does not apply to the ring excitation geometry. In fact, in a ring excitation geometry polaritons from next-neighbours will not be blocked by the repulsive potential of each adjacent spot, and as a result all pump spots will be coupled to each other with different coupling strengths. If that's the case, the ring geometry is by no means equivalent to Fig. 3b with periodic boundary conditions. I have previously asked the authors about the appearance of gaps in the ring geometry. They have added a new sentence to the manuscript: "We point out that for our detection setup the extraction of polariton band features along the polygons circumference in Fig. 6 is not possible at current". While this is understandable from their new experiment setup figure (e.g. absence of a stepper motor to allow for a tomographic acquisition of 2-dimensional energy dependencies), I don't think a full band picture along the whole circumference is necessary to observe an energy gap. For example, one can simply acquire a 1-D energy dependence along a line intersecting two pump spots belonging to the ring, which should show a gap if the author's claims are true. According to the setup image, exciting with a linear or with a ring geometry does not change any of the acquisition part, so nothing prevents the authors from taking spectrometer data while changing their excitation profile. Can the authors provide such a data? Even if that data was not or cannot be taken, theoretical simulations should be able to clarify this discussion. Full 1-dimensional, or even full 2-dimensional, energy profiles can be produced with their simulated data. Could the authors provide such a simulation and, hopefully, show that a gap is indeed present?

The referee correctly points out the presence of radial modes in Fig.6 which result in observable interference across the polygon's centre and thus finite interaction beyond nearest neighbour. We however point out that Fig.6 is plotted in a logarithmic color-scale and the strongest interference (derived from the intensity of fringes between condensates) is clearly between nearest neighbours.

Figure 1 in this reply shows the same data that is presented in Fig.6 of the manuscript but in a linear colour scale to stress this further.

Figure 1: Experimental polariton condensate PL using eight pump spots forming a circle. Linear colourmaps showing the (a & c) real-space and (b & d) k-space photoluminescence distributions where (a & b) and (c & d) show regular and dimerised octagons respectively. The black lines are $15 \mu\text{m}$ and $1 \mu\text{m}^{-1}$ scale bars in (a & c) and (b & d) respectively.

The polygon does not possess translational symmetry, on this point we agree, it does however possess discrete rotational symmetry. This means that Bloch theorem can still be applied along the circumcircle of the system regardless of radial effects (since particle eigenmodes must also commute with the symmetry group of the system). These single-particle modes satisfy $\psi_{q,n}(\mathbf{r}) = e^{iq\varphi} u_{q,n}(\mathbf{r})$ where $u_{q,n}(r, \varphi) = u_{q,n}(r, \varphi + 2\pi/N)$ is the Bloch mode of the system and $q = 2\pi m/L$ is the angular quasi-momentum along the polygon's circumcircle of length L and N is the number of lattice unit cells. By projecting the circumcircle of the polygon onto a 1D coordinate one can calculate the energies of the angular Bloch mode wavenumbers which we demonstrate in Fig.2 in this letter and in the modified Fig.2 in the supplemental material.

Figure 2: Numerically resolved dispersion of a uniform and staggered octagon respectively along its circumcircle showing gap opening of the discrete wavenumbers q corresponding to angular particle propagation.

The ring geometry introduces some weak hopping terms also beyond the nearest neighbour owing to the small size of the ring. The coupling strength between the condensates decreases exponentially as a function of the separation distance, so the effect of the next-nearest-neighbour represents a much weaker correction compared with the coupling of the nearest neighbour. All such effects become even weaker when the polygon's radius increases.

We also emphasise that we have based the analysis of the systems largely on the tight-binding approximation that already assumes that various higher-order band effects could be ignored along the direct coupling between the condensates. In small rings, the validity of the tight-binding approximation may become even more limited, but as long as the general features of the couplings can be described by such an approximation the concept of periodic boundary conditions remains valid. The tight-binding approximation could also be improved by including the next-nearest-neighbour as a small correction, without changing any of the conclusions of the calculations. We now stress this in the manuscript where we have added the following sentence,

“The presence of very weak radial modes clearly seen in logarithmic colour scale in Fig. 6 can be minimized by increasing the polygon's size. As long as the general features of the couplings between the condensates can be approximated by the tight-binding model, the assumption of the periodic boundary conditions remains valid.”

2) The authors have kept the discussion about Zak phases in the main text, however in a simplified version to improve readability – which is much appreciated. However, they have failed to give the background of such a topic, and especially how it is important for the data described here. Stating that the concept is a “very hot topic” and giving numerous references to it does not help one understand the meaning of it. Therefore I must ask again: which role does Zak phases play into the data displayed here? Is it related to the opening of the gap, just from topological continuity arguments? For example, will the size of the energy gap be given by a condition where the wavefunction phase undergoes a $0, 2\pi, 4\pi$ etc phase jump from one end to the other end of the Brillouin zone? Does it help to give topological protection to any of the observable quantities? I insist, the way the whole discussion around Zak phases is presented in the text seems to be a pure mathematical exercise. The text still fails to provide any intuition about how it influences the observed experimental data.

The opening of the gap in our system (experimentally shown in Fig.2 and calculated in Fig.3) is associated with a phase transition where the system goes from being topologically trivial (uniform lattice) to topologically non-trivial (staggered lattice). It is not possible at the current to measure the Zak phase directly through interferometric techniques in our polariton system like demonstrated in previous works on cold atoms using a combination of Bloch oscillations with Ramsey interferometry [Atal et al., Nat. Phys. 9, 795 (2013)] or by realising a Thouless pump [Nakajima et al., Nat. Phys. 12, 296 (2016)]. Such manipulation of polariton wavepackets (or condensates) is beyond the scope of our work. The reason the gap opening implies a topological phase transition comes directly from the localization of polariton modes at each pump spot. It is analogous to particles occupying a single mode at each site in the lowest band of deep periodic potentials (i.e., wavefunction is a superposition of localized Wannier functions). This is why we believe non-Hermitian systems can

open up new avenues towards topological physics since, as we demonstrate, the localization of the particles is not bound to the potential minima of a lattice with tunnelling dictated by evanescent tunnelling rates.

In order to make this point clearer to the reader, we have added the following sentence:

“Experimentally, the gap opening observed in Fig. 2 implies a topological phase transition due to the localization of polariton modes at each pump spot. This is in analogy to deep periodic potentials where the particles occupy a single mode at each site in the lowest band (i.e., the wavefunction can be described as a superposition of localised Wannier functions). The strong non-Hermitian nature of our hybrid light-matter system opens new avenues towards topological physics since, where the localisation of the particles is not dictated by the potential minima of the lattice with evanescent tunnelling.”

I thank the authors for explaining that the simulated dispersions are for single particles, therefore not to a condensate, and apologize for not having noticed it. However, I am now confused: Fig. S1i shows that the polariton single-particle dispersion reflects the lattice’s periodicity, and Fig. S1l shows that dimerization opens up gaps in the single-particle dispersion. Also, according to Figs. S5c,fi,l, soliton-like defect states also seem to be present in the single-particle physics they present. Therefore, one naturally wonders if condensation is necessary to observe any of the main features described in this paper, namely gaps and solitonic defects. In their response to referee #3, the authors seem to state that this is indeed the case: “The band structure features we observe are visible immediately from reaching threshold laser power and remain even notably above threshold”, Could the authors present such a experimental data where, for the same pump spot geometry, just before condensation threshold no gaps nor solitonic features are present, however they appear above condensation? If that’s the case, could the authors explain why, in response to referee #3, the authors say that “one-particle plots show what accessible dispersion looks like to polaritons prior to condensation”?

We thank the referee for a valuable remark. The sentence “*The band structure features we observe are visible immediately from reaching threshold laser power and remain even notably above threshold*” only says that the bands become visible after condensation which is just a matter of having enough polaritons in the system to occupy these bands. The bands also exist in the linear regime (i.e., below threshold) and would be observable by either shining a broadband white light or by resonantly scanning the system in frequency.

Following the sentence “*one-particle plots show what accessible dispersion looks like to polaritons prior to condensation*” to referee #3 in previous reply letter we also wrote “*Our calculations are performed for low pump powers such that nonlinear renormalization of the single-particle dispersion due to the presence of the condensate is negligible*”. In other words, we expect that the dispersion which we see in experiment above threshold is approximately the same as felt by single polaritons below threshold. This goes in hand with the study’s objectives to show that the optically engineered dispersion in the linear regime will still apply to the condensate above threshold, long as the characteristic on-site interaction energy,

$$U = \alpha \int |\Psi(\mathbf{r})|^2 d\mathbf{r}$$

evaluated over the lattice unit cell, is smaller than the engineered band gap and band width of the lattice. The defect state lasing/condensation observed in experiment is also not expected to renormalize the particle dispersion dramatically due to the low powers which we operate at. We feel that the terminology of *dark gap solitons* and *bright gap solitons* is justified by the self-trapping polariton condensates experience by expelling particles from their pumped region in-order to stabilise [Ostrovskaya et al., Phys. Rev. A 86, 013636 (2012)]. However, there exists in principle a family of solitonic solutions by varying, for instance, the particle number of the condensate in the defect state which was not the subject of the study. As such, we believe our observations and demonstration of the system flexibility are better delivered by adjusting our statement on *dark gap solitons* and *bright gap solitons* to a more general case. We have adjusted our introductory sentence starting “Finally, by introducing local defects...” to the following,

“Finally, by introducing local defects in the potentials periodicity, we demonstrate controllable highly localised defect state condensation opening up possibilities to investigate analogues of bright and dark solitonic gap modes in strongly non-Hermitian lattices.”

We have also adjusted the following sentence,

“Such states are the optically generated analog of polariton bright gap solitons observed in [39]. On the other hand, the delocalised band energetically above the defect state suffers significant suppression in condensate occupation spatially around the defect, representing a dark soliton mode (see Fig.4d).”

to

“Such strongly localized states could permit investigation into optically generated analogue of polariton bright gap solitons observed previously for polariton condensates in photonic lattices [39]. On the other hand, the delocalised band energetically above the defect state suffers significant suppression in condensate occupation spatially around the defect, representing a dark soliton-like mode (see Fig.4d).”

4) I thank the authors for adding ticks to the logarithmic colour scales. However, I must ask them to also add labels to the ticks in the main text data, or at least say in which basis the logarithmic scale is. As it is now, one cannot know if, e.g., the ratio between black and grey is 10, e, 2, 100, etc.

All the figures have been replotted and now include labels on the colourbars.

5) Apart from now stating that the exciton-photon detuning is -3.5meV, could they please also state what the sample’s Rabi splitting detuning is, such that one knows the photonic/excitonic character of the presented data?

In the methods sections, the sentence describing the detuning has been adapted to also include the vacuum Rabi splitting as was determined in ref. 48 of the manuscript.

6) In the caption of Fig S1: “Panels (c,f)” should read “Panels (i,l)” instead.

Typo has been fixed now.

REVIEWERS' COMMENTS:

Reviewer #2 (Remarks to the Author):

I believe the authors have addressed all my comments satisfactory. I especially thanks them for the addition of Supp. Fig. 2c,f, which certainly highlights and helps understand the radially periodic nature of the pump spots. May I ask them to please correct the figure labels accordingly? (a,c->a,d; b,d->b,e).

I recommend the paper for publication.

We thank all the reviewers for the time they have taken to provide constructive criticism throughout the review process.

Reviewer 2.

I believe the authors have addressed all my comments satisfactory. I especially thanks them for the addition of Supp. Fig. 2c,f, which certainly highlights and helps understand the radially periodic nature of the pump spots. May I ask them to please correct the figure labels accordingly? (a,c->a,d; b,d->b,e).

I recommend the paper for publication.

We thank reviewer 2 for bringing this to our attention, all references to the Supplementary Figure 2 have been corrected.